# ShapeMatch: Shapelet-Guided Semi-Supervised Learning for Multivariate Time Series Classification

## Abstract

Multivariate Time Series Classification (MTSC) is crucial for many real-world applications and deep learning models such as Transformer have become the state-of-the-art (SOTA) for MTSC due to their ability to capture complex temporal and spatial dependencies. However, they struggle to perform well without sufficient labelled data, limiting their effectiveness in label-scarce scenarios. Furthermore, the absence of effective augmentation methods for time series data that can enhance generalisation whilst preserving essential temporal structures poses a significant challenge. As a result, despite the success of semi-supervised learning in other domains, these limitations have left its integration with deep learning-based MTSC largely unexplored. To bridge this gap, we propose ShapeMatch, a novel flexible semi-supervised framework designed to enhance MTSC in label-constrained environments. ShapeMatch introduces two key innovations: (1) a hybrid training approach that leverages the classic Shapelet Model to guide the deep learning model in the early stages, capitalising on Shapelets' robustness for low-label regimes, and (2) ShapeAug, a tailored augmentation strategy for multivariate time series that preserves critical structural patterns whilst introducing meaningful variations. Extensive experiments on benchmark datasets demonstrate that ShapeMatch surpasses existing SOTA methods for scenarios with limited labelled data, making it a powerful solution for real-world MTSC applications. Our code is available at http://anonymous.4open.science/r/Shape-Match-MTSC/.

## 1 Introduction

Multivariate Time Series Classification (MTSC) is a pivotal area of research in machine learning, driven by its extensive applications in domains such as healthcare, finance, and industrial monitoring Stevner et al. (2019); Ruiz et al. (2021a); Patton (2012); Ruiz et al. (2021b). In this context, deep learning models like Transformer-based architectures Vaswani et al. (2017); Dosovitskiy et al. (2020); Devlin et al. (2018) or Convolution-based architectures Eldele et al. (2024) have emerged as the state-of-the-art (SOTA) for MTSC Le et al. (2024); Zhou et al. (2023a); Wu et al. (2021); Liu et al. (2023a); Wang et al. (2024) due to their ability to model intricate temporal dependencies and capture complex interactions across multiple variables. However, their effectiveness heavily depends on the availability of large-scale labelled datasets, making them less viable in real-world scenarios where annotated data is often scarce. The high cost and domain expertise required for manual labelling further exacerbate this challenge. For example, in healthcare, labelling heartbeat, ECG, or EEG data is expensive due to the involvement of medical experts Zhai et al. (2020). Similarly, domains such as human activity recognition and IoT also require domain expertise for annotation Yue et al. (2022), further motivating the need for semi-supervised learning.

Semi-supervised learning (SSL) Sohn et al. (2020); Weng et al. (2022); Li et al. (2021b) has recently emerged as a promising approach to overcome label scarcity by leveraging labelled and unlabelled instances during training. Although such methods have been applied to time series analysis Liu et al. (2024; 2023b); Wei et al. (2023), most of them focus solely on feature-based univariate time series classification and are not applicable to multivariate time series, where deep learning based architectures (DL) now represent the state-of-the-art.

The limited exploration of semi-supervised learning for deep-learning-based MTSC can be attributed to the following challenges:

**(1) Deep-learning-based MTSC are highly sensitive to label scarcity:** Despite their ability to model complex temporal and spatial structures, deep learning models such as Transformer struggle to perform well without sufficient labelled data. In fact, these models yield significantly poorer performance for label-scarce settings (as shown in Figure 1). This often results in significantly degraded performance and unreliable pseudo-labels when applied in semi-supervised frameworks like FixMatch Sohn et al. (2020), CoMatch Li et al. (2021b), MixMatch Berthelot et al. (2019) or its variants.

**(2) Lack of effective MTSC augmentation:** Although data augmentation has been highly successful and is a crucial component in domains, such as computer vision Sohn et al. (2020); Li et al. (2021b), MTSC lacks effective augmentation methods that enhance generalisation, while preserving essential temporal patterns.

To address these challenges, we introduce **Shape-Match** (Shapelet-Guided Matching), a novel semi-supervised framework that bridges the strengths of

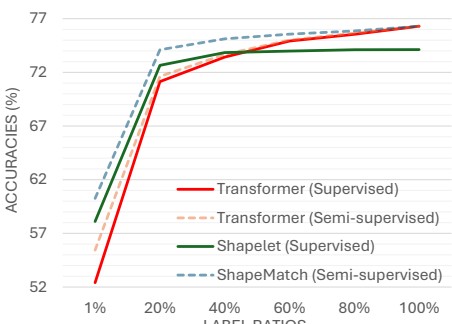

Figure 1: Accuracy of Transformer (Supervised), Transformer (Semi-Supervised using FixMatch), Shapelet Model (Supervised), and ShapeMatch (Semi-Supervised) on APAVA datasets. Shapelet Models perform better at lower label ratios, Transformer excels with more labels. By proposing a hybrid approach ShapeMatch that leverages the Shapelet Model to guide the Transformer in the early stages, we can achieve high accuracy across most label ratio settings.

both classic and modern approaches to multivariate time series classification. ShapeMatch incorporates two key innovations: (1) a hybrid training paradigm that leverages the Shapelet Model Ye & Keogh (2009); Le et al. (2022), known for its robustness in low-label regimes (as demonstrated in Figure 1), to guide the DL model in *matching* the predictions of the Shapelet Model during early training; and (2) ShapeAug, a specialised augmentation technique for multivariate time series that preserves essential structural patterns while introducing meaningful variations. By incorporating shapelet-based guidance, the DL model learn more effective information during early learning, while ShapeAug enhances the model's ability to extract useful representations from unlabeled data. As shown in Figure 1, ShapeMatch achieves high accuracy across most label ratio settings.

Extensive experiments on twelve benchmark datasets demonstrate that ShapeMatch not only achieves SOTA performance in semi-supervised scenarios. Our findings highlight the effectiveness of Shapelet-based guidance in DL architectures and emphasise the importance of tailored augmentation in MTSC. By addressing both data efficiency and augmentation challenges, ShapeMatch represents a significant step forward in advancing multivariate time series analysis.

Our key contributions are summarised as follows:

- We propose **ShapeMatch**, a novel semi-supervised framework for multivariate time series classification (MTSC) that integrates Shapelet-based guidance into deep learning models, enabling more effective learning during the early stages of training.
- ShapeMatch is a versatile framework that is highly compatible with, and performs effectively on, various transformer-based and convolution-based backbones.
- We introduce **ShapeAug**, a tailored augmentation strategy for multivariate time series that preserves essential structural patterns while introducing meaningful variations, enabling the model to extract richer information from unlabeled data during semi-supervised learning.
- We perform extensive experiments on benchmark datasets, demonstrating that ShapeMatch achieves SOTA performance in semi-supervised settings, establishing its effectiveness in real-world applications.

## 2 RELATED WORK

**Multivariate Time Series Classification.** MTSC methods fall into two main categories: traditional classic models and more recent deep learning (DL) models. Classic models are typically compact and include approaches like Dynamic Time Warping with 1-Nearest Neighbour Berndt & Clifford (1994); Keogh & Ratanamahatana (2005), time series shapelets Ye & Keogh (2009); Li et al. (2021a); Le et al. (2022), which extract discriminative subsequences for each class, and other feature-based methods Dempster et al. (2021); Zhang et al. (2020). Recently, DL based model such as convolution-based model Eldele et al. (2024) and transformer-based models have achieved state-of-the-art (SOTA) performance in time series classification Wang et al. (2024); Le et al. (2024). GTN

Liu et al. (2021) employs a two-tower multi-headed attention mechanism, whilst ConvTran Foumani et al. (2023) enhances position embeddings with absolute and relative encoding. SVP-T Zuo et al. (2023) uses clustering to identify subsequences, improving long- and short-term dependency modelling. ShapeFormer Le et al. (2024) integrates shapelets for better performance, and MedFormer Wang et al. (2024) applies Transformers to healthcare time series. Successful transformer models rely on large labelled datasets, limiting their real-world applicability where annotated data is scarce. In contrast, classic methods, especially shapelet-based models Le et al. (2022; 2024), are more efficient and perform well in low-label settings.

**Univariate vs Multivariate Time Series Semi-Supervised Learning.** Semi-supervised learning (SSL) Sohn et al. (2020); Weng et al. (2022); Li et al. (2021b) has emerged as a powerful paradigm for mitigating label scarcity by leveraging both labelled and unlabelled data, which is particularly important in real-world settings where annotation is costly or infeasible. By enhancing representation learning and improving generalisation, SSL effectively bridges the gap between supervised and unsupervised learning. While several SSL methods have been proposed for univariate Liu et al. (2024; 2023b); Wei et al. (2023) and multivariate Du et al. (2025) time series classification, they typically introduce task-specific semi-supervised architectures Du et al. (2025) that are less flexible and difficult to apply on top of state-of-the-art backbones such as Transformer-, CNN-, or LLM-based time series models. In parallel, self-supervised approaches Eldele et al. (2023); Yue et al. (2022) have also been explored; however, their downstream performance remains highly sensitive to label scarcity, especially during the early supervised adaptation stage. Moreover, most deep SSL methods critically rely on strong data augmentation strategies, which are still relatively underdeveloped for time series data.

## 3 PRELIMINARIES

**Multivariate Time Series Classification (MTSC):** We represent MTS as $\mathbf{X} \in \mathbb{R}^{V \times T}$, where $V$ denotes the number of variables and $T$ represents the length of the time series. Here, $\mathbf{X} = \{\mathbf{X}^1, \ldots, \mathbf{X}^V\}$, and each $\mathbf{X}^v$ corresponds to a time series for variable $v$. Note $\mathbf{X}^v = \{x^{v,1}, \ldots, x^{v,T}\}$, where $x^{v,t}$ signifies a value for variable $v$ at timestamp $t$ within $\mathbf{X}$. Consider a training dataset $\mathcal{D} = \{(\mathbf{X}_i, y_i)\}_{i=1}^{M}$, where $M$ is the number of time series instances, $\mathcal{Y}$ are the label sets ($y_i \in \mathcal{Y}$) and $|\mathcal{Y}|$ is the number of classes, and the pair $(\mathbf{X}_i, y_i)$ represents a training sample and its corresponding label, respectively. The objective of MTSC is to train a classifier $\mathcal{F}(\mathbf{X})$ to predict a class label for a multivariate time series with an unknown label.

**Semi-Supervised Learning for MTSC (MTSC-SSL):** In this setting, we consider two datasets: a labelled dataset $\mathcal{D}_L = \{(\mathbf{X}_i, y_i)\}_{i=1}^{M_L}$ and an unlabelled dataset $\mathcal{D}_U = \{\mathbf{X}_i\}_{i=1}^{M_U}$, where $M_L + M_U = M$. The goal of MTSC-SSL is to train a model $\mathcal{F}$ with high accuracy by leveraging information from both $\mathcal{D}_L$ and $\mathcal{D}_U$.

## 4 PROPOSED METHODS

We propose **ShapeMatch**, a **novel flexible SSL framework** that integrates the strengths of both classic and modern approaches to enhance the performance of various time-series classification backbone in semi-supervised settings. The training process consists of four key stages. First, in the **Shapelet Model Initialisation** (Section 4.1), class-specific discriminative shapelets are extracted from the labelled dataset and used to initialise the Shapelet Model $\mathcal{F}_S$. Next, **Augmentation with ShapeAug** (Section 4.2) applies transformations to both labelled and unlabelled samples, enhancing generalisation. In the **Labelled Dataset Pre-training** stage (Section 4.3), both the Shapelet Model $\mathcal{F}_S$ and the DL backbone $\mathcal{F}_T$ are trained on the labelled dataset $\mathcal{D}_L$. Finally, during **Semi-Supervised Training** (Section 4.4), the DL backbone learns using a combination of labelled loss, Shapelet-guided regularisation, and pseudo-labelling on unlabelled data.

By integrating Shapelet-based guidance for DL model learning and a robust augmentation strategy, ShapeMatch effectively increase significantly the performance of the DL models in semi-supervised setting. The overall architecture is shown in Figure 3.

### 4.1 SHAPELET MODEL INITIALISATION

**Shapelet.** In 2009, Ye et al. introduced shapelets, discriminative subsequences of time series, for classification Ye & Keogh (2009). Shapelets effectively capture local patterns that outperform global trends in distinguishing between classes, while also offering interpretable classification decisions. Their effectiveness has been demonstrated for univariate and multivariate time series classification tasks Le et al. (2024; 2022).

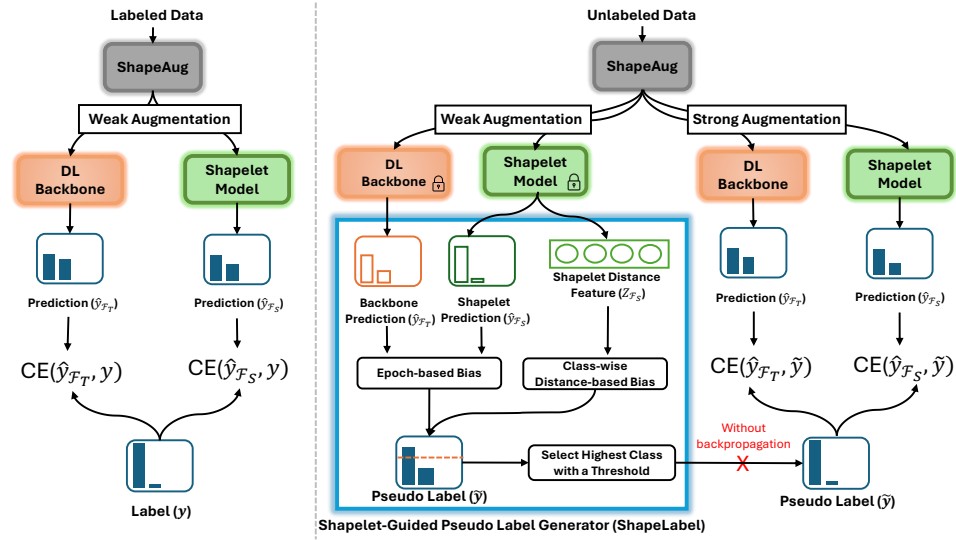

Figure 2: The architecture of Shapelet-Guided Semi-Supervised Learning leverages the strengths of DL model and Shapelet Model. (a) Initially, both models are trained on labelled data with a weak shapelet-guided augmentation. (b) Next, unlabelled data is augmented using ShapeAug to generate weak and strong augmentations. Pseudo-labels are generated from weakly augmented data using frozen DL model and Shapelet Models (i.e., with parameters locked during training) without backpropagation (red cross). This process incorporates both epoch-based and class-wise distance-based bias. These pseudo-labels are then used to train the models on strongly augmented data, enhancing learning through semi-supervised guidance. It is important to note that **our method is a framework compatible with any deep learning backbone.**

**Shapelet Discovery.** Our framework starts by extracting class-discriminative shapelets from labelled data using a novel **Perceptual and Position-aware Shapelet Discovery (PPSD)** method for multivariate time series. Inspired by PPSN and ShapeFormer Le et al. (2022; 2024), PPSD leverages *Perceptually Important Points (PIPs)* Chung et al. (2001) to efficiently identify compact and informative shapelet candidates. Unlike traditional methods, PPSD drastically reduces candidate volume and computation. Shapelets are ranked by information gain, and the top ones are stored in a shapelet pool $\mathcal{S}$ for model training.

To ensure efficiency, we sample only $r = 50$ time series per class (compared to using all samples in prior work). Despite this, our ablation studies (**Appendix E**) confirm that PPSD maintains competitive performance while significantly improving speed. Full extraction and selection details are provided in the Algorithm 1 and **Appendix A** sections.

**Adapted to Multivariate Time Series:** In contrast to shapelet discovery method used in PPSD Le et al. (2022) and ShapeFormer Le et al. (2024), which uses only univariate shapelets, our Shape-Match leverages both univariate (within one channel) and multivariate shapelets (over all channels) to better capture inter-variable dependencies.

**Shapelet Model.** Given the shapelet pool $\mathcal{S} = \{S, y^{\text{shape}}\}_{i=1}^{g*|\mathcal{Y}|}$, where $S$ is a shapelet, $y_i^{\text{shape}}$ is the class label for shapelet $S_i$ and $g * |\mathcal{Y}|$ is the total number of selected shapelets, we follow the PPSN model Le et al. (2022) to initialise the shapelet model $\mathcal{F}_S$ using $\mathcal{S}$ as initial weights. Then, for each $\mathbf{X} \sim \mathcal{D}$, the shapelet distance features $\boldsymbol{Z}_{\mathcal{F}_S} = \{z\}_{i=1}^{g}$ are computed by applying the shapelet-distance Le et al. (2022) to all shapelets $S \in \mathcal{D}$.

$$z_i = \text{ShapeletDistance}(\mathbf{X}, S_i), \tag{1}$$

where, ShapeletDistance follows the definition from Le et al. (2022); Ye & Keogh (2009) and quantifies the minimum distance between the shapelet $S_i$ and any subsequence of equal length in the time series $\mathbf{X}$.

After that, $\boldsymbol{Z}_{\mathcal{F}_S}$ is normalised and then fed into a simple neural network containing a ReLU activation function and a single linear layer.

$$\hat{y}_{\mathcal{F}_S} = \text{argmax}(\text{softmax}(\text{Linear}(\text{ReLU}(\boldsymbol{Z}_{\mathcal{F}_S})))) . \tag{2}$$

Please note that all shapelets $S \in \mathcal{S}$ are learnable using the cross-entropy loss function $L_{CE}$.

## 4.2 ShapeAug: Shapelet-Guided Augmentation for Multivariate Time Series

Traditional time series augmentation methods struggle to balance class-specific feature preservation with meaningful variability. Excessive transformations can distort key discriminative patterns, reducing classification performance. To address this challenge, we introduce **ShapeAug**, an augmentation strategy that selectively modifies time series while ensuring that critical class-defining features remain intact.

**Best-Matching Subsequences.** ShapeAug first identifies **best-matching subsequence (B)**, which is a subsequence within a target time series that has the smallest distance to each selected shapelet. These positions highlight the most class-representative segments of the series. Using these identified regions, ShapeAug applies controlled augmentation techniques that preserve class-specific structures whilst introducing diversity in other aspects of the time series.

**Shapelet-Guided Mask.** The mask $\mathcal{M} = [m^{1,1}, \ldots, m^{V,T}] \in \mathbb{R}^{V \times T}$ is calculated as follows:

$$m^{v,t} = \begin{cases} \text{PSD}(\mathbf{B}', \mathbf{S}') & \text{if } \exists \mathbf{B}' \in \mathbf{B}, \text{where } x^{v,t} \in \mathbf{B}', \\ 1 & \text{otherwise}, \end{cases} \tag{3}$$

where $\mathbf{S}'$ is the corresponding shapelet for best-matching subsequence $\mathbf{B}'$.

Given the time series instances $\mathbf{X} = [x^{1,1}, \ldots, x^{V,T}] \in \mathcal{D}$ of class $Y \in \mathcal{Y}$, the shapelet-guided mask $\mathcal{M} = [m^{1,1}, \ldots, m^{V,T}]$, and the augmentation scale $\sigma$, ShapeAug consists of four key augmentation techniques:

**(a) Random Jittering:** Introduces small random noise to the time series, which helps enhance model robustness whilst preserving the underlying structure. To maintain the class-specific information, the impact of this technique is minimised at the shapelet positions.

$$\tilde{\mathbf{X}} = \mathbf{X} + \mathcal{E}^{\text{Jitter}} \odot \mathcal{M}, \tag{4}$$

where $\mathcal{E}^{\text{Jitter}} = [\epsilon^{1,1}, \ldots, \epsilon^{V,T}] \sim \mathcal{N}(0, \sigma^2)$ represents Gaussian noise, and $\odot$ denotes element-wise multiplication.

If $x^{v,t} \notin \mathbf{B}$, noise is added as $\tilde{x}^{v,t} = x^{v,t} + \epsilon^{v,t}$. Otherwise, noise is scaled: $\tilde{x}^{v,t} = x^{v,t} + \text{PSD}(\mathbf{B}', \mathbf{S}') \cdot \epsilon^{v,t}$. A small $\text{PSD}(\mathbf{B}', \mathbf{S}')$ keeps $\mathbf{X}$ almost unchanged. Using the shapelet-guided mask, this process ensures that the augmented instances are different from the original data while still retaining important shape characteristics specific to their class.

**(b) Random Masking:** Randomly sets certain points to zero, simulating missing data. As with Random Jittering, we minimise the modifications at shapelet positions to preserve class-specific information, ensuring that the key features remain intact in the time series.

$$\tilde{\mathbf{X}} = \mathbf{X} \odot \mathcal{E}^{\text{Mask}}, \tag{5}$$

where $\mathcal{E}^{\text{Mask}}_{i,j} \sim \text{Bernoulli}\left(1 - \exp\left(-\frac{1}{\sigma m^{i,j}}\right)\right)$. This formula ensures that a higher $\sigma$ and a lower value in $\mathcal{M}$ lead to a higher probability of masking the position.

**(c) Shapelet-Scaling + Cropping:** Crops the time series whilst focusing on the shapelet positions, then scales the remaining sequence up or down to introduce temporal variations.

**(d) Random Shifting:** Shifts the time series left or right by a random window size, maintaining structural integrity while encouraging generalisation.

**Weak and Strong Shapelet-Guided Augmentations:** To further control augmentation strength, we propose two variants of ShapeAug:

- **Weak Shapelet-Guided Augmentation (WeakSAug):** Combines only *Random Jittering* and *Random Shifting* with small scaling factors, introducing mild variations while keeping the time series structure largely intact.
- **Strong Shapelet-Guided Augmentation (StrongSAug):** Applies all four augmentation techniques with higher scaling factors, generating diverse yet class-consistent variations of the time series.

By leveraging shapelet positions to control augmentation intensity, ShapeAug effectively enhances data diversity while preserving essential classification features. This balance makes it especially well-suited for semi-supervised learning, where maintaining feature integrity alongside sufficient augmentation is critical for model performance, thereby enabling the model to extract richer information from unlabeled data.

### 4.3 Labelled Dataset Pre-training

Given the selected shapelets $\mathcal{S}$, we initialise the Shapelet Model $\mathcal{F}_S$ to capture discriminative features and the DL model $\mathcal{F}_T$ to learn long-range temporal dependencies. Our framework is compatible with any deep learning backbone, including Convolution-based TSLANet Eldele et al. (2024)

and Transformer Vaswani et al. (2017), iTransformer Liu et al. (2023a), ShapeFormer Le et al. (2024), MedFormer Wang et al. (2024), PatchTST Nie et al. (2022), and others.

For a labelled data pair $(X, y) \sim \mathcal{D}_L$, we generate augmented data $\tilde{X}^w$ using Weak Shapelet-guided Augmentation (Section 4.2):

$$\tilde{\mathbf{X}}^w = \text{WeakSAug}(\mathbf{X}). \tag{6}$$

We then train the models using cross-entropy loss, simultaneously optimising both models on the labelled dataset $\mathcal{D}_L$:

$$\mathcal{L}_L = \mathcal{L}_{CE}(\hat{y}_{\mathcal{F}_S}, y) + \mathcal{L}_{CE}(\hat{y}_{\mathcal{F}_T}, y), \tag{7}$$

where $\hat{y}_{\mathcal{F}_S} = \mathcal{F}_S(\tilde{\mathbf{X}}^w)$ and $\hat{y}_{\mathcal{F}_T} = \mathcal{F}_T(\tilde{\mathbf{X}}^w)$.

This training phase enables the models to learn discriminative shapelet features and long-range temporal dependencies, improving the generation of pseudo-labels for the semi-supervised learning phase on unlabelled data.

### 4.4 Shapelet-Guided Semi-Supervised Learning

Given pre-trained $\mathcal{F}_S$ and $\mathcal{F}_T$ from Labelled Dataset Pre-training and unlabelled dataset $\mathcal{D}_U$, we propose Shapelet-Guided Semi-Supervised Learning to further train $\mathcal{F}_S$ and $\mathcal{F}_T$ with the information from $\mathcal{D}_U$. The key idea is to guide the learning process using the previously extracted shapelets to impose constraints on the model learning, ensuring that the learnt features remain aligned with meaningful temporal patterns. By doing so, we can effectively use the unlabelled data to enhance model generalisation without needing additional labelled data.

Given $\mathbf{X} \sim \mathcal{D}_U$, inspired by various semi-supervised frameworks Sohn et al. (2020); Li et al. (2021b); Weng et al. (2022), we generate the weak and strong augmented versions of $\mathbf{X}$ using Shapelet-guided Augmentation (Section 4.2).

$$\tilde{\mathbf{X}}^w = \text{WeakSAug}(\mathbf{X}), \quad \tilde{\mathbf{X}}^s = \text{StrongSAug}(\mathbf{X}) \tag{8}$$

After that, $\tilde{\mathbf{X}}^w$ is then fed into frozen $\mathcal{F}_T$ and $\mathcal{F}_S$ (i.e., with parameters locked during training).

$$\tilde{y}^w_{\mathcal{F}_T} = \mathcal{F}_T(\tilde{\mathbf{X}}^w), \tag{9}$$

$$\tilde{y}^w_{\mathcal{F}_S}, \mathbf{Z}^w_{\mathcal{F}_S} = \mathcal{F}_S(\tilde{\mathbf{X}}^w), \tag{10}$$

where $\mathbf{Z}^w_{\mathcal{F}_S}$ is the shapelet distance feature of model $\mathcal{F}_S$ for $\tilde{\mathbf{X}}^w$. Please note that both $\tilde{y}^w_{\mathcal{F}_T}$ and $\tilde{y}^w_{\mathcal{F}_S}$ are the labels before softmax normalisation.

**Shapelet-Guided Pseudo Label Generator (ShapeLabel):** In this component, we propose methods to generate pseudo-labels for unlabelled data using both the Shapelet Model $\mathcal{F}_S$ and the DL model $\mathcal{F}_T$. From this, the pseudo-label $\tilde{y} \in \mathbb{R}^{1 \times |\mathcal{Y}|}$ for $\mathbf{X}$ (where $|\mathcal{Y}|$ is the number of classes) is generated using the following two biases:

*Epoch-based Bias:* In this approach, we propose using more of the shapelet model predictions and fewer DL model predictions in the early stages, gradually reducing this bias in the later stages of training.

$$\tilde{y} = \lambda_e \tilde{y}^w_{\mathcal{F}_S} + (1 - \lambda_e)\tilde{y}^w_{\mathcal{F}_T}, \tag{11}$$

where $\lambda_e$ is a time-dependent weight function that starts with higher values during the early stages and gradually decreases as training progresses.

$$\lambda_e = \frac{1}{2}\left(1 + cos\left(\frac{\pi e}{e_{\max}}\right)\right), \tag{12}$$

where $e$ is the current epoch and $e_{\max}$ is the total number of epochs. From that we can achieve this:

- **Early-Stage Stability:** The pseudo-label $\tilde{y}$ provides a strong inductive bias towards $\tilde{y}^w_{\mathcal{F}_S}$ during the early stage, ensuring the learning of meaningful features despite the limited availability of labelled samples.
- **Late-Stage Adaptability:** In the later stage, the pseudo-label $\tilde{y}$ reduces the bias towards $\tilde{y}^w_{\mathcal{F}_S}$ and increases the bias towards $\tilde{y}^w_{\mathcal{F}_T}$, enabling the DL model to generalise better as more data becomes available.

Table 1: Performance comparison of our proposed ShapeMatch model against supervised learning and two representative semi-supervised methods, TS2VEC Yue et al. (2022), CA-TTC Eldele et al. (2023), one semi-supervised specific model semiHGR Du et al. (2025), and four semi-supervised framework: Pseudo-Label Lee et al. (2013), FixMatch Sohn et al. (2020), and Semiformer Weng et al. (2022), using the Classic Transformer backbone Vaswani et al. (2017) across five healthcare benchmark datasets Wang et al. (2024): APAVA, TD-Brain, ADFTD, PTB, and PTB-XL. The supervised results on the full dataset (upper-bound accuracy with 100% labels) are obtained from Wang et al. (2024).

| Category | Method | APAVA (76.30%) | | | TDBrain (87.17%) | | | ADFTD (50.47%) | | | PTB (77.37%) | | | PTB-XL (70.59%) | | |
|---|---|---|---|---|---|---|---|---|---|---|---|---|---|---|---|---|
| | | 1% | 5% | 20% | 1% | 5% | 20% | 1% | 5% | 20% | 1% | 5% | 20% | 1% | 5% | 20% |
| Supervised-Learning | Supervised | 52.41 | 63.52 | 71.14 | 60.21 | 68.41 | 78.25 | 41.24 | 43.06 | 45.21 | 57.36 | 66.28 | 73.32 | 52.34 | 60.97 | 65.71 |
| Self-Supervised Learning | TS2VEC | 54.12 | 64.98 | 72.32 | 62.92 | 70.87 | 79.43 | 42.54 | 44.36 | 46.51 | 59.07 | 67.74 | 74.11 | 54.05 | 62.43 | 66.29 |
| Self-Supervised Learning | CA-TTC | 55.06 | 65.77 | 72.87 | 64.36 | 72.01 | 80.02 | 42.64 | 44.46 | 46.61 | 59.51 | 67.88 | 74.32 | 54.29 | 62.72 | 67.06 |
| Semi-SL Model | semiHGR | 55.61 | 65.92 | 72.93 | 64.41 | 72.16 | 80.10 | 42.66 | 44.51 | 46.66 | 59.56 | 67.93 | 74.47 | 54.39 | 62.82 | 67.01 |
| Semi-SL Framework | Pseudo-Label | 54.12 | 65.13 | 72.19 | 61.86 | 69.71 | 79.91 | 41.68 | 43.72 | 45.98 | 59.06 | 67.91 | 74.13 | 53.49 | 61.93 | 66.31 |
| Semi-SL Framework | FixMatch | 55.44 | 66.52 | 72.42 | 63.16 | 71.11 | 80.56 | 42.12 | 44.12 | 46.41 | 60.16 | 68.42 | 74.24 | 54.31 | 63.36 | 66.42 |
| Semi-SL Framework | Semiformer | 55.31 | 65.92 | 72.65 | 62.21 | 72.72 | 81.98 | 41.97 | 44.32 | 46.28 | 59.55 | 68.98 | 74.12 | 54.47 | 63.83 | 66.54 |
| Semi-SL Framework | ShapeMatch | **60.26** | **69.12** | **74.11** | **67.72** | **77.24** | **84.14** | **44.21** | **46.12** | **48.11** | **65.14** | **71.23** | **75.79** | **58.44** | **65.89** | **68.62** |

*Class-wise Distance-based Bias:* In this state, we propose to use the shapelet distance features $Z^s_{\mathcal{F}_S}$ (Equation 1) to further refine the pseudo-labels $\tilde{y}$. First, we calculate the averaged distance features $\bar{Z}_{\mathcal{F}_S} = \{\bar{z}\}_{i=1}^{|\mathcal{Y}|}$ for each class. Then $\bar{Z}$ is normalised such that the class with the lowest distance (i.e., the most similar class) is assigned a value of 1, while the other classes are assigned values less than 1.

$$\bar{z}_i = \frac{\bar{z}_i - \min(\bar{Z})}{\max(\bar{Z}) - \min(\bar{Z})} \quad \text{for} \quad i = 1, 2, \ldots, |\mathcal{Y}|. \tag{13}$$

After that, the pseudo-label $\tilde{y}$ is multiplied by $\bar{Z}$ to normalise the values and apply the softmax function to generate the final pseudo-label.

$$\tilde{y} = \text{softmax}(\tilde{y} \times \bar{Z}). \tag{14}$$

**Select Highest Class with a Threshold:** Finally, inspired by Sohn et al. (2020), we use a threshold $\tau$ for generating pseudo-labels to ensure that the prediction confidence is assessed, and the pseudo-label is only assigned if the confidence exceeds this threshold.

$$\tilde{y} = \begin{cases} \text{argmax}(\tilde{y}) & \text{if } \max(\tilde{y}) \geq \tau, \\ \text{no pseudo-label} & \text{otherwise.} \end{cases} \tag{15}$$

**Shapelet-Guided Strong-Augmented Data Learning:** After generating the pseudo-label $\tilde{y}$ using the weakly augmented version $\tilde{X}^w$ of $X$, we use the pseudo-label $\tilde{y}$ to train both $\mathcal{F}_T$ and $\mathcal{F}_S$ on strongly augmented data $\tilde{X}^s$.

$$\mathcal{L}_U = \mathcal{L}_{CE}(\hat{y}^s_{\mathcal{F}_S}, \tilde{y}) + \mathcal{L}_{CE}(\hat{y}^s_{\mathcal{F}_T}, \tilde{y}), \tag{16}$$

where, $\hat{y}^s_{\mathcal{F}_S} = \mathcal{F}_S(\tilde{X}^s)$ and $\hat{y}^s_{\mathcal{F}_T} = \mathcal{F}_T(\tilde{X}^s)$.

The use of pseudo-labels $\tilde{y}$ from weakly augmented data $\tilde{X}^w$ for training on strongly augmented data $\tilde{X}^s$ provides a stable learning foundation. Strong augmentation can cause overfitting due to data variation, but weak augmentation helps maintain consistency across transformations, reducing noisy predictions and improving model robustness. Additionally, guiding the DL model with the Shapelet Model during early training leverages the strengths of both methods. Shapelet Models capture discriminative subsequences, providing a strong inductive bias that helps the deep learning model learn key patterns in label-scarce environments, enhancing training performance.

## 4.5 OVERALL FRAMEWORK

Our framework, **ShapeMatch**, begins by extracting shapelets from the labelled dataset to initialise the Shapelet Model $\mathcal{F}_S$, while the DL backbone $\mathcal{F}_T$ is randomly initialised (our ShapeMatch supports any DL backbone). Next, ShapeAug is applied to augment the labelled data, which is then used to train both models. In the semi-supervised stage, the pre-trained models continue training with unlabelled data using Shapelet-guided semi-supervised learning. Unlabelled data undergoes StrongSAug and WeakSAug, where weakly augmented samples are passed through the frozen models to extract predictions and shapelet features. These features are then processed by a Shapelet-guided pseudo-label generator to create pseudo-labels, which are used to train strongly augmented data. After training, only the DL backbone is used for inference. By integrating Shapelet-based guidance, DL model learning, and a robust augmentation strategy, ShapeMatch significantly improve the performance of DL backbone in Semi-Supervised MTSC settings.

Table 2: Performance comparison of our proposed ShapeMatch model against four popular semi-supervised framework, one semi-supervised specific model and two self-supervised method using Classic Transformer Backbone Vaswani et al. (2017) on seven UEA datasets Bagnall et al. (2018). The supervised results on the full dataset (upper bound accuracy, using 100% labels) are obtained from Le et al. (2024).

| Category | Method | CharacTraject (99.60%) | | | FaceDetection (63.25%) | | | LSST (61.60%) | | | Phoneme (29.30%) | | | SpokenAraD (99.30%) | | | PenDigits (98.40%) | | | InsectWing (65.80%) | | |
|---|---|---|---|---|---|---|---|---|---|---|---|---|---|---|---|---|---|---|---|---|---|---|---|
| | | 1% | 5% | 20% | 1% | 5% | 20% | 1% | 5% | 20% | 1% | 5% | 20% | 1% | 5% | 20% | 1% | 5% | 20% | 1% | 5% | 20% |
| Supervised-Learning | | 72.33 | 82.06 | 90.70 | 40.02 | 45.97 | 54.68 | 52.30 | 53.71 | 57.11 | 13.77 | 23.75 | 26.00 | 73.41 | 81.74 | 90.56 | 78.24 | 83.24 | 86.56 | 47.51 | 51.25 | 53.91 |
| Self-Supervised Learning | TS2VEC | 74.12 | 85.11 | 93.89 | 41.10 | 47.80 | 55.10 | 52.70 | 54.00 | 56.90 | 14.60 | 24.40 | 26.30 | 74.00 | 82.40 | 91.10 | 79.10 | 84.70 | 87.90 | 48.80 | 52.40 | 54.70 |
| Self-Supervised Learning | CA-TTC | 75.71 | 86.45 | 95.12 | 41.80 | 50.10 | 55.50 | 52.60 | 55.00 | 56.90 | 14.90 | 25.10 | 27.10 | 75.10 | 83.80 | 91.60 | 80.10 | 85.10 | 88.30 | 49.00 | 53.20 | 55.80 |
| Semi-SL Model | semiHGR | 74.51 | 86.52 | 95.23 | 42.00 | 50.40 | 55.80 | 53.00 | 55.50 | 57.00 | 15.10 | 25.30 | 27.30 | 76.20 | 84.30 | 91.80 | 80.80 | 85.80 | 88.60 | 49.60 | 53.50 | 56.00 |
| Semi-SL Framework | Pseudo-Label | 74.07 | 86.90 | 93.90 | 40.34 | 49.29 | 54.30 | 51.76 | 54.14 | 56.78 | 14.35 | 25.66 | 26.58 | 74.22 | 82.79 | 91.46 | 79.89 | 85.58 | 88.76 | 49.58 | 53.31 | 56.16 |
| Semi-SL Framework | FixMatch | 74.25 | 88.28 | 92.70 | 42.27 | 49.90 | 56.69 | 54.02 | 54.23 | 57.49 | 15.62 | 24.20 | 27.79 | 75.37 | 83.53 | 91.42 | 80.67 | 85.69 | 88.77 | 49.95 | 53.79 | 56.81 |
| Semi-SL Framework | Semiformer | 73.46 | 86.21 | 95.63 | 42.10 | 51.79 | 56.36 | 52.95 | 55.80 | 57.05 | 15.09 | 25.51 | 27.63 | 77.01 | 84.94 | 92.14 | 81.56 | 86.26 | 89.66 | 50.49 | 53.89 | 56.83 |
| Semi-SL Framework | ShapeMatch | **78.60** | **90.96** | **97.84** | **47.35** | **55.13** | **58.29** | **55.14** | **57.84** | **59.58** | **18.76** | **26.23** | **28.67** | **79.54** | **86.55** | **93.04** | **85.21** | **93.45** | **96.24** | **54.53** | **57.53** | **60.01** |

Table 3: Accuracy comparison of ShapeMatch with different backbones across varying label ratios on APAVA dataset. ShapeMatch consistently achieves the highest accuracy across all settings.

| | TSLANet Eldele et al. (2024) (74.21%) | | | iTransformer Liu et al. (2023a) (74.55%) | | | ShapeFormer Le et al. (2024) (79.25%) | | | MedFormer Wang et al. (2024) (78.84%) | | | PatchTST Nie et al. (2022) (74.55%) | | |
|---|---|---|---|---|---|---|---|---|---|---|---|---|---|---|---|
| Label Ratio | 1% | 5% | 20% | 1% | 5% | 20% | 1% | 5% | 20% | 1% | 5% | 20% | 1% | 5% | 20% |
| Supervised Learning | 52.16 | 62.82 | 66.64 | 50.57 | 62.11 | 67.75 | 58.26 | 69.52 | 72.41 | 56.02 | 66.47 | 72.65 | 51.56 | 54.64 | 57.53 |
| Pseudo-Label | 54.41 | 64.71 | 69.08 | 51.38 | 62.42 | 69.78 | 58.51 | 71.82 | 73.32 | 56.14 | 67.84 | 73.85 | 52.64 | 55.21 | 60.34 |
| FixMatch | 54.18 | 65.26 | 69.02 | 52.07 | 65.06 | 68.99 | 61.32 | 71.91 | 74.42 | 57.78 | 69.79 | 73.72 | 53.66 | 57.42 | 59.86 |
| Semiformer | 55.58 | 65.77 | 69.91 | 53.03 | 63.37 | 68.74 | 60.45 | 72.64 | 73.47 | 58.54 | 68.10 | 73.99 | 53.91 | 56.24 | 60.54 |
| ShapeMatch | **57.09** | **66.41** | **71.49** | **57.05** | **67.42** | **71.97** | **65.35** | **74.48** | **75.92** | **62.73** | **71.85** | **74.92** | **57.92** | **60.75** | **63.03** |

## 5 EXPERIMENTS

### 5.1 EXPERIMENTAL SETTINGS

**Dataset.** We selected five widely used healthcare time series datasets Wang et al. (2024) to demonstrate the practical benefits of our approach, and seven datasets from the UEA archive Bagnall et al. (2018). It is important to note that SSL requires a sufficient amount of labelled data for meaningful evaluation; however, most UEA datasets contain fewer than 500 labeled samples. Therefore, we limited our selection to the seven datasets that meet this criterion to effectively showcase the performance of our method. Full details of all datasets are provided in **Appendix C**.

**Baselines.** Upper-bound supervised results were taken from Wang et al. (2024); Le et al. (2024). We also compared our method with four semi-supervised approaches: (1) Supervised Learning using available labelled data, (2) Pseudo-Label Lee et al. (2013), (3) FixMatch Sohn et al. (2020), and (4) Semiformer Liu et al. (2023b), the first semi-supervised method for vision transformers.

**Implementation Details.** In all experiments, we split the training set into labelled and unlabelled subsets using **label ratios of 1%, 5%, and 20%**. Our model was trained with the RAdam optimiser (learning rate 0.01, momentum 0.9, weight decay 5e-4) for 200 epochs ($e_{max}$) with a batch size of 16. Results were averaged over three random seeds (1, 10, 100) to ensure robustness. Accuracy, following the protocol in Sohn et al. (2020); Weng et al. (2022), was used as the main metric.

### 5.2 PERFORMANCE EVALUATION

**Healthcare Time Series Datasets.** Table 1 shows that ShapeMatch consistently outperforms all baselines across label ratios (1%, 5%, 20%). On TDBrain, it achieves 84.14% accuracy, exceeding the best baseline (Semiformer) by 2.16% at 20%. Similar gains are observed on APAVA and PTB, where ShapeMatch surpasses the strongest baselines by 4–6% under low-label regimes, highlighting its effectiveness in leveraging limited labelled data.

**UEA Datasets.** Table 2 further demonstrates ShapeMatch's consistent superiority across diverse datasets and label ratios. On CharacterTrajectories, it reaches 97.84% accuracy, outperforming Semiformer by 2.21% at 20%. Substantial improvements are also seen on FaceDetection (+1.93%) and LSST (+2.53%), while in low-label settings, ShapeMatch achieves a 3.67% gain on Phoneme at 1%. These results confirm its robustness and effectiveness for semi-supervised MTSC.

### 5.3 COMPARISON WITH DIFFERENT BACKBONES

To further evaluate the effectiveness of ShapeMatch, we assess its performance across four different backbone architectures: iTransformer Liu et al. (2023a), ShapeFormer Le et al. (2024), MedFormer Wang et al. (2024), and PatchTST Nie et al. (2022). The accuracy comparisons, presented in Table 3, show that ShapeMatch consistently outperforms all baselines across varying label ratios (1%, 5%, and 20%). In addition, we evaluate its performance with the CNN-based TSLANet model Eldele et al. (2024). The results indicate that ShapeMatch still significantly outperforms other SSL methods

Table 4: Left: Component evaluation across two datasets, APAVA and FaceDetection, at varying label ratios (1%, 5%, and 20%). Right: Accuracies of our ShapeMatch with ablation for each augmentation strategy of ShapeAug.

| Dataset | APAVA (76.30%) | | | FaceDetection (63.25%) | | |
|---|---|---|---|---|---|---|
| Label Ratio | 1% | 5% | 20% | 1% | 5% | 20% |
| FixMatch | 55.44 | 66.52 | 72.42 | 42.27 | 49.90 | 56.69 |
| + ShapeAug | 56.30 | 67.61 | 73.39 | 44.13 | 52.41 | 57.45 |
| + ShapeLabel | 57.03 | 67.68 | 73.67 | 45.53 | 54.50 | 57.52 |
| + ShapeAug + ShapeLabel | **60.26** | **69.12** | **74.11** | **47.35** | **55.13** | **58.29** |

| Dataset | APAVA (76.30%) | | | FaceDetection (63.25%) | | |
|---|---|---|---|---|---|---|
| Label Ratio | 1% | 5% | 20% | 1% | 5% | 20% |
| FixMatch | 55.44 | 66.52 | 72.42 | 42.27 | 49.90 | 56.69 |
| Without Random Jittering | 58.67 | 66.85 | 72.22 | 45.84 | 53.19 | 55.92 |
| Without Random Masking | 58.38 | 66.78 | 71.96 | 45.49 | 52.98 | 55.95 |
| Without Shapelet-Scaling + Crop | 59.04 | 67.36 | 72.62 | 45.44 | 54.06 | 57.10 |
| Without Random Shifting | 59.00 | 67.62 | 72.48 | 46.11 | 53.59 | 56.72 |
| ShapeMatch | **60.26** | **69.12** | **74.11** | **47.35** | **55.13** | **58.29** |

(a) Our ShapeAug    (b) Traditional Random Jittering

Figure 3: Comparison between our ShapeAug (a) and traditional Random Jittering (b). The red segments indicate shapelet positions, which capture the essential class-discriminative information. ShapeAug preserves these critical subsequences during augmentation, while traditional random jittering may distort them.

when paired with the CNN architecture, further demonstrating its adaptability and generalisability across different deep learning models for MTSC. We also conducted the experiment with other backbone like LLM-based model Zhou et al. (2023b) in **Appendix G**.

### 5.4 Ablation Study

**Component Evaluation.** We begin by evaluating the impact of key components on shapelet initialisation: ShapeAug (Section 4.2), and ShapeLabel (Section 4.4). As shown in Table 4 (left), introducing ShapeAug results in modest improvements across both datasets, while adding ShapeLabel further enhances accuracy. However, the combination of ShapeAug and ShapeLabel consistently yields the highest performance, achieving the best results in all cases. This demonstrates the effectiveness of augmenting and labelling shapelets together, particularly at higher label ratios, where the improvements are more pronounced.

**Effectiveness of ShapeAug.** We performed an ablation study to evaluate the impact of using ShapeAug's difference augmentation strategies. As shown in Table 4 (right), removing individual ShapeAug components results in varying degrees of performance degradation across both datasets. While each augmentation technique contributes to improved accuracy, their combined effect in ShapeMatch consistently yields the best results in all cases, especially at lower label ratios.

We present additional results in **Appendix**.

## 6 Visualization

Figure 3 presents a comparison between ShapeAug and traditional Random Jittering. ShapeAug preserves the essential patterns while introducing meaningful variations, whereas Random Jittering tends to distort the signal. This highlights ShapeAug's capacity to maintain semantic integrity while simultaneously enhancing data diversity for training.

## 7 Conclusion

In this paper, we propose ShapeMatch, a novel semi-supervised framework for multivariate time series classification that incorporates Shapelet-based guidance into deep learning models, enhancing learning efficiency especially during early training stages. We also introduce ShapeAug, a specialized augmentation technique designed to preserve critical structural patterns in multivariate time series while injecting meaningful variability, enabling more effective utilization of unlabeled data. Our framework demonstrates strong compatibility and robust performance across diverse transformer- and convolution-based architectures.

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

## A    FULL DETAIL FOR SHAPELET DISCOVERY

---

**Algorithm 1:** Perceptual and Position-aware Shapelet Discovery

---

**Input:** $\mathcal{D}$: dataset; time series length $T$; channels $V$; number of PIPs $k$; number of shapelets $g$ per class; classes $\mathcal{Y}$ with $|\mathcal{Y}|$ as the number of classes.

1 For each class $Y \in \mathcal{Y}$, reduce its samples to $r$, meaning only $r$ samples per class are used in the subsequent process. $\mathcal{S} = []$ # Shapelet sets;                                                                    #
2 **foreach** $(\mathbf{X}, y) \in \mathcal{D}$ **do**
3    **for** $v = 1$ *to* $V$ **do**
4       $\boldsymbol{P} = [1, T]$ # Initialise PIPs set;                                                                    #
5       **for** $j = 1$ *to* $k - 2$ **do**
6          Identify index $p$ that maximises reconstruction distance;
7          Insert $p$ into $\boldsymbol{P}$, sort, and determine its index $idx$;
8          **for** $z = 0$ *to* 2 **do**
9             **if** $idx + 2 - z \leq |\boldsymbol{P}|$ *and* $idx - z \geq 1$ **then**
10               $s\_pos = \boldsymbol{P}[idx - z], e\_pos = \boldsymbol{P}[idx + 2 - z]$;
              # For univariate shapelets                                                                    #
11               Extract univariate candidate $S =$ from $\mathbf{X}[s\_pos : e\_pos]$ within channel $V$ and append it into $\mathcal{S}$;
              # For multivariate shapelets                                                                    #
12               Extract multivariate candidate $S$ from $\mathbf{X}[s\_pos : e\_pos]$ with all channels and append it into $\mathcal{S}$;
13 **foreach** $(S, y) \in \mathcal{S}$ **do**
14    Compute the information gain of $S$ for class $y$ using Eq. 1 with all $\mathbf{X}_i \in \mathcal{D}$;
15 **foreach** $\tilde{Y} \in \mathcal{Y}$ **do**
16    Select the top $g$ candidates $S \in \mathcal{S}$ of class $\tilde{Y}$ by information gain and discard the rest.;
17 **return** $\mathcal{S}$

---

Our framework begins by extracting discriminative shapelets from the labelled dataset. i.e. key subsequences that capture class-specific patterns. To achieve this efficiently, we introduce the Perceptual and Position-aware Shapelet Discovery (PPSD) method, inspired by PPSN and ShapeFormer Le et al. (2022; 2024), for multivariate time series. PPSD utilises Perceptually Important Points (PIPs)

Chung et al. (2001) to identify crucial points based on reconstruction distance, enabling precise and compact shapelet extraction. Unlike traditional methods, PPSD significantly reduces computational overhead by generating far fewer candidates. Finally, shapelets are ranked by their information gain, with the most informative ones stored in the shapelet pool $\mathcal{S}$ for model training.

PPSD operates in two main phases: shapelet extraction and shapelet selection, as outlined in Algorithm 1. First, it identifies shapelet candidates by selecting PIPs. The process starts by including the first and last indices in the PIPs set, then iteratively adding the index with the highest reconstruction distance. Each newly added PIP can generate up to three shapelet candidates using consecutive PIPs. In the selection phase, PPSD ensures an equal number of $g$ shapelets per class. For each candidate $S_i$ belonging to class $y$, its Perceptual Subsequence Distance Le et al. (2022) is computed against all training instances, as defined in Eq. 1.

**Complexity Analysis:** The overall computational complexity of the shapelet discovery phase is based on the number of channels $V$, time-series length $T$, number of time series $N$, and number of selected shapelets $g$. This phase consists of two stages: *shapelet candidate discovery* and *shapelet selection*. In the candidate discovery stage, we search approximately $0.2NVT$ salient points to obtain $g$ candidates. In the selection stage, we compute the information gain of each candidate over all $N$ time series, giving a complexity of $\mathcal{O}(gN^2D)$, where $D$ is the cost of computing the distance between one shapelet and one time series (with $D = T$ when using PISD as the distance). Summarising both stages, the overall complexity is

$$\mathcal{O}\big(0.2VNT + gNT\big). \tag{17}$$

## B  PIP: RECONSTRUCTION DISTANCE

Assume a time series $Q = [q_1, q_2, \ldots, q_n]$ and let $k$ be the number of key points to extract. We begin by adding the first and last indices $(1, n)$ to the PIP list. Then, we recursively select the index with the largest perpendicular distance from the line connecting two existing PIPs.

The perpendicular distance of a point $z$ with respect to a line defined by two PIPs is computed as:

$$PD(z, \text{PIPs}) = \frac{|a \cdot P_z - Q_z + c|}{\sqrt{a^2 + 1}},$$

where

$$a = \frac{Q_e - Q_s}{P_e - P_s}, \qquad c = Q_e - a \cdot P_e.$$

Here, $s = \text{PIPs}_g$ and $e = \text{PIPs}_{g+1}$, given some $g$ where $1 \leq g \leq k$ and $\text{PIPs}_g < z < \text{PIPs}_{g+1}$. Finally, let $P$ denote the list of positions after z-normalization:

$$P = z\_norm([1, \ldots, n]).$$

**Adapted to MTSC:** Our method extracts both univariate (within a single channel) and multivariate (across all channels) shapelets to better capture information for multivariate time series classification.

In contrast to PPSN and ShapeFormer Le et al. (2022; 2024), our method uses a fixed number of $r = 50$ time series per class (compared to using all samples as PPSN and ShapeFormer), which significantly accelerates the shapelet initialisation process. Our ablation study (**Appendix E**) shows that despite using a smaller number of time series, the performance of our method remains comparable, while substantially speeding up the process.

## C  DATASET

**APAVA:** The APAVA dataset Wang et al. (2024) is a public EEG dataset with 23 subjects (12 Alzheimer's patients, 11 healthy controls) recorded across 16 channels. Each 5-second trial (1280 timestamps) is standardised and segmented into 1-second samples (256 timestamps), yielding 5,967 samples. A subject-independent split assigns validation (15,16,19,20), test (1,2,17,18), and the rest to training.

**TDBrain:** The TDBrain dataset Wang et al. (2024) includes EEG recordings from 1274 subjects with 33 channels. We use a subset of 50 subjects (25 Parkinson's patients, 25 healthy controls) in eyes-closed condition. Each trial is segmented into 1-second samples (256 timestamps), resulting

Table 5: Statistics of datasets.

|  | Dataset | Training Size | Test Size | Channels | Length | Classes |
|---|---|---|---|---|---|---|
| Healthcare | APAVA | 3580 | 716 | 16 | 256 | 2 |
| | ADFTD | 41851 | 8370 | 19 | 256 | 3 |
| | TDBrain | 3744 | 749 | 33 | 256 | 2 |
| | PTB | 38614 | 7723 | 15 | 300 | 2 |
| | PTB-XL | 114840 | 22968 | 12 | 250 | 5 |
| UEA | CharacterTrajectories | 1422 | 1436 | 3 | 182 | 20 |
| | FaceDetection | 5890 | 3524 | 144 | 62 | 2 |
| | LSST | 2459 | 2466 | 6 | 36 | 14 |
| | Phoneme | 3315 | 3353 | 11 | 217 | 39 |
| | SpokenArabicDigits | 6599 | 2199 | 13 | 93 | 10 |
| | PenDigits | 7494 | 3498 | 2 | 8 | 10 |
| | InsectWingbeat | 30000 | 20000 | 200 | 78 | 10 |

in 6,240 samples. A subject-independent split assigns validation (18,19,20,21,46,47,48,49), test (22,23,24,25,50,51,52,53), and the rest to training.

**ADFTD:** The ADFTD dataset Wang et al. (2024) is a public EEG dataset with 88 subjects (36 Alzheimer's, 23 Frontotemporal Dementia, 29 healthy controls) recorded across 19 channels at 500 Hz. Trials are bandpass-filtered (0.5–45 Hz), downsampled to 256 Hz, and segmented into 1-second samples (256 timestamps), yielding 69,752 samples. Both subject-dependent and subject-independent splits allocate 60%, 20%, and 20% of samples/subjects to training, validation, and testing.

**PTB:** The PTB dataset Wang et al. (2024) is a public ECG dataset with 290 subjects, 15 channels, and 8 labels (7 heart diseases, 1 healthy control). We use a subset of 198 subjects (Myocardial infarction and healthy controls). Signals are downsampled to 250 Hz, normalised, and segmented into single heartbeats using R-Peak detection, yielding 64,356 samples. A subject-independent split assigns 60%, 20%, and 20% of subjects to training, validation, and testing.

**PTB-XL:** The PTB-XL dataset Wang et al. (2024) is a large public ECG dataset with 18,869 subjects, 12 channels, and 5 labels (4 heart diseases, 1 healthy control). To ensure consistency, we retain 17,596 subjects with uniform diagnoses. The 500 Hz signals are downsampled to 250 Hz, normalised, and segmented into 1-second samples (250 timestamps), resulting in 191,400 samples. A subject-independent split allocates 60%, 20%, and 20% of subjects to training, validation, and testing.

**UEA Datasets:** We follow the default setting in Bagnall et al. (2018) and use them for all experiments.

Details of these datasets are provided in Table 5.

# D COMPUTATIONAL RESOURCE COMPARISON

We provide a detailed comparison of memory usage, GPU VRAM, and training time over three methods, including ShapeMatch, a supervised Transformer, and FixMatch with a Transformer backbone during both training and inference, as shown in Table 6. All experiments were performed on a single Intel(R) Xeon(R) Silver 4214 CPU @ 2.20 GHz and one NVIDIA Tesla V100 SXM2 GPU. While our method introduces a slight increase in resource usage, specifically an additional 12 MB of memory, 0.5 GB of VRAM, and 0.63 hours of training time compared to FixMatch, this overhead is minimal and well justified. The added cost primarily stems from the shapelet discovery and integration process, which is essential to the performance improvement. This represents a favourable trade-off, with modest training overhead yielding significant gains. All these additional components are deactivated during inference, allowing real-time operation without any extra cost at test time.

Table 6: Resource comparison during training and inference.

| Method | Memory Usage | GPU VRAM (Max) | Training Time | Inference Time |
|---|---|---|---|---|
| Supervised | NA | 2.8GB | 0.72h | 11.4s |
| FixMatch | NA | 6.2GB | 1.04h | 11.4s |
| ShapeMatch | 12MB | 6.7GB | 1.07h + 0.6h | 11.4s |

**Computation Cost of Shapelet Discovery:** The CPU-based computation may raise some concern; however, we assert that CPU-based shapelet computation is not a practical bottleneck. On 8 CPU cores, it took about 36 minutes, but with modern cloud servers (e.g., Google Cloud), this can be reduced to under 1 minute at a minimal cost of $0.368 (see Table 7). This demonstrates that shapelet computation is fast, inexpensive, and not a limiting factor in real applications.

Table 7: Shapelet discovery computation settings and cost.

| | CPU | Cores Used | RAM | Discovery Time | Cost | Note |
|---|---|---|---|---|---|---|
| Our Experiment Setting | Intel(R) Xeon(R) Silver 4214 | 8 | 64 GB | $\approx$36 min | N/A | |
| Recommendation | Google Cloud (c4-highmem-288) | 288 | 2232 GB | <1 min | $0.368 | $22.1 per hour |

# E  SENSITIVITY ANALYSIS OF HYPERPARAMETERS

**Different Numbers of Time Series Per Class ($r$) Used for Shapelet Initialisation.** We conducted experiments to analyse the impact of varying the number of time series used for shapelet initialisation. As shown in Table 8, the highest accuracy for all label ratios is consistently achieved at $r = 50$, with the performance remaining stable for larger values of $r$. Notably, using a larger number of time series, such as $r = 100$, $r = 200$, or the full dataset, does not significantly improve accuracy, but increases the shapelet initialisation time substantially. For instance, at $r = 50$, the shapelet initialisation time is 1.1 hours, whilst for the full dataset, it rises to 10.5 hours. This demonstrates a trade-off between running time and accuracy, where $r = 50$ provides a good balance.

Table 8: Accuracies and running time for various time series $r$ used for Shapelet Initialisation.

| No. of Time Series $r$ | | 10 | 30 | 50 | 100 | 200 | Full |
|---|---|---|---|---|---|---|---|
| | Label Ratio 1% | 60.31 | 60.23 | **60.26** | 60.26 | 60.22 | 60.24 |
| APAVA | Label Ratio 5% | 69.13 | 69.12 | **69.12** | 69.06 | 69.05 | 69.03 |
| (76.30%) | Label Ratio 20% | 74.13 | 74.11 | **74.11** | 74.12 | 74.12 | 74.11 |
| | Shapelet Init Time | 0.3h | 0.8h | **1.1h** | 2.6h | 4.5h | 10.5h |

**Different Number of Selected Shapelets $g$.** We conducted experiments to analyse the impact of varying the window size and the number of shapelets on classification accuracy. As shown in Table 9, the highest accuracy is achieved when the window size is set to 50 and the number of shapelets is 30, reaching **69.12%**. Increasing the number of shapelets beyond this point does not yield substantial improvements in accuracy. Similarly, smaller window sizes generally result in lower accuracy, indicating that an appropriate choice of window size is crucial for optimal performance. These findings highlight a trade-off between computational cost and classification accuracy, where selecting an optimal combination of window size and shapelets is essential for achieving best results.

Table 9: Accuracies for various values of window size and number of shapelets in Shapelet Discovery.

| Window size \ Shapelets | 1 | 3 | 10 | 30 | 100 |
|---|---|---|---|---|---|
| 10 | 66.10 | 66.40 | 66.56 | 68.10 | 68.52 |
| 20 | 65.78 | 66.81 | 66.64 | 67.57 | 68.18 |
| 50 | 65.81 | 66.45 | 66.53 | **69.12** | 68.82 |
| 100 | 65.28 | 67.1 | 66.62 | 68.05 | 68.2 |
| 200 | 65.28 | 66.84 | 66.94 | 67.17 | 68.9 |

**Different Augmentation Scaling Factor $\sigma$.** We performed experiments to assess how varying the augmentation scaling factor $\sigma$ affects classification accuracy across different label proportions. As shown in Table 10, accuracy generally improves as $\sigma$ increases, reaching the highest values at $\sigma = 0.8$ across all label proportions. Specifically, at 1% labelled data, accuracy peaks at **60.26%**, while at 5% and 20% labelled data, the best accuracies are **69.12%** and **74.11%**, respectively. Beyond this optimal point, further increasing $\sigma$ does not yield significant improvements. These results suggest that introducing an appropriate level of noise can enhance model performance, but excessive noise may lead to diminishing returns.

Table 10: Accuracies for various values of augmentation scaling factor $\sigma$ in ShapeAug.

| Scaling factor $\sigma$ | | 0.1 | 0.2 | 0.4 | 0.6 | 0.8 | 1 | 2 |
|---|---|---|---|---|---|---|---|---|
| | 1% | 56.45 | 58.00 | 57.68 | 58.38 | **60.26** | 59.62 | 59.99 |
| APAVA | 5% | 65.72 | 67.11 | 66.93 | 68.12 | **69.12** | 68.94 | 69.04 |
| (76.30%) | 20% | 71.05 | 71.29 | 72.00 | 72.67 | **74.11** | 73.16 | 73.17 |

**Different Decay Methods for $\lambda_e$ (Equation 12).** We compare different methods for decaying the value of $\lambda_e$ in Equation 12, including Step Decay, where $\lambda_e$ is halved every 50 steps; Linear Decay, which reduces $\lambda_e$ according to a linear formula; and our selected Cosine Decay, as described in Equation 12. As shown in Table 11, Cosine Decay consistently outperforms the other decay

methods across both datasets, APAVA and FaceDetection, at all label ratios. Specifically, Cosine Decay achieves the highest accuracies, with notable improvements over FixMatch, Step Decay, and Linear Decay. These results highlight the effectiveness of Cosine Decay in enhancing performance, especially at higher label ratios, where it provides a significant boost in classification accuracy.

Table 11: Left: Accuracy with various decay methods for ShapeLabel.

| Dataset | APAVA (76.30%) | | | FaceDetection (63.25%) | | |
|---|---|---|---|---|---|---|
| Label Ratio | 1% | 5% | 20% | 1% | 5% | 20% |
| FixMatch | 55.44 | 66.52 | 72.42 | 42.27 | 49.90 | 56.69 |
| Step Decay | 56.80 | 68.42 | 73.25 | 44.04 | 51.52 | 57.25 |
| Linear Decay | 58.02 | 70.20 | 73.67 | 45.87 | 53.01 | 57.81 |
| Cosine Decay | **60.26** | **69.12** | **74.11** | **47.35** | **55.13** | **58.29** |

**Threshold $\tau$ (Equation 15):** We analyse the effect of varying $\tau$ on ShapeMatch performance using the APAVA dataset, as shown in Table 12. Accuracy improves with higher $\tau$ values, peaking at $\tau = 0.9$ across all label ratios. At 1%, 5%, and 20% label ratios, the highest accuracies are 60.26%, 69.12%, and 74.11%, respectively, demonstrating the effectiveness of higher thresholds.

Table 12: Accuracy of ShapeMatch across different threshold values $\tau$ on APAVA.

| Threshold $\tau$ | | 0.5 | 0.6 | 0.7 | 0.8 | 0.85 | 0.9 | 0.95 |
|---|---|---|---|---|---|---|---|---|
| APAVA | 1% | 56.35 | 57.27 | 57.40 | 58.93 | 59.26 | **60.26** | 59.50 |
| (76.30%) | 5% | 65.92 | 66.69 | 66.30 | 67.63 | 68.70 | **69.12** | 68.59 |
| | 20% | 70.77 | 71.35 | 71.62 | 73.05 | 73.35 | **74.11** | 73.99 |

**Experiments with Different Types of Small Models:** We conducted additional experiments using a small CNN model and a MiniRocket model as the backbone. These results further clarify that our shapelet-based model still achieves better performance across different small backbone architectures.

Table 13 shows the performance comparison on both the APAVA and FaceDetection datasets under different label ratios. Our shapelet-based model consistently outperforms the alternative small models, demonstrating the effectiveness of the shapelet guidance.

Table 13: Performance comparison with different small backbone models on APAVA and FaceDetection datasets.

| Model | APAVA | | | FaceDetection | | |
|---|---|---|---|---|---|---|
| | 1% | 5% | 20% | 1% | 5% | 20% |
| Shapelet Model (Default) | **60.26** | **69.12** | **74.11** | **47.35** | **55.13** | **58.29** |
| MiniRocket | 56.34 | 66.74 | 72.12 | 43.51 | 51.92 | 57.41 |
| 3-layer CNN | 55.11 | 65.23 | 71.95 | 43.14 | 51.83 | 57.32 |

**Ablation Study for Epoch-based Bias and Class-wise Distance-based Bias:** We conducted an ablation study to analyse the effect of Epoch-based Bias and Class-wise Distance-based Bias in our ShapeMatch framework. The results for both the APAVA and FaceDetection datasets are shown in Table 14.

Table 14: Ablation study on the effect of Epoch-based Bias and Class-wise Distance-based Bias.

| Dataset | Method | 1% | 5% | 20% |
|---|---|---|---|---|
| APAVA | ShapeMatch | **60.26** | **69.12** | **74.11** |
| | − Epoch-based Bias | 57.21 | 68.01 | 73.62 |
| | − Class-based Distance-based Bias | 58.92 | 68.67 | 73.81 |
| | − Epoch-based & Class-based Bias | 56.30 | 67.61 | 73.39 |
| FaceDetection | ShapeMatch | **47.35** | **55.13** | **58.29** |
| | − Epoch-based Bias | 45.61 | 53.12 | 57.71 |
| | − Class-based Distance-based Bias | 46.45 | 54.07 | 57.92 |
| | − Epoch-based & Class-based Bias | 44.13 | 52.41 | 57.45 |

As observed, removing either component consistently decreases performance, and removing both leads to the largest drop. This demonstrates that both the epoch-based bias and the class-based distance-based bias make important contributions to the overall effectiveness of ShapeMatch.

# F  FURTHER DISCUSSION

## F.1  PROBLEM OF PRETRAINING DL BACKBONE

We observed that pretraining the deep learning (DL) backbone and subsequently using it for pseudo-label generation can further improve performance. This improvement can be attributed to the fact that, during pretraining on labeled data, the DL backbone acquires background knowledge of the data distribution. Consequently, it does not rely solely on the guidance of shapelets in the early stages, which allows the backbone to learn more effectively and converge faster. For instance, with a label ratio of 20%, a non-pretrained backbone would lack exposure to these 20% of the dataset, limiting its generalization ability.

On the other hand, using the DL backbone to generate pseudo-labels in the early stage introduces a potential risk of overfitting. However, this risk is mitigated by the use of an **Epoch-Based Bias**, whereby the contribution (weight) of these early-stage pseudo-labels remains very small, serving primarily as a weak auxiliary signal rather than a dominant influence.

To demonstrate this claim, we conducted an ablation study comparing the performance when (i) the DL backbone is not pretrained, (ii) pseudo-labels from the DL backbone are not used, and (iii) both are removed simultaneously. The results are shown in Table 15.

Table 15: Ablation study on the effect of pretraining the DL backbone and using DL backbone pseudo-labels.

| Method | 1% | 5% | 20% |
|---|---|---|---|
| ShapeMatch | **60.26** | **69.12** | **74.11** |
| − Pretrained DL Backbone | 57.48 | 66.72 | 71.15 |
| − DL Backbone's Pseudo Label | 56.97 | 66.54 | 70.81 |
| − Pretrained DL Backbone & Pseudo Label | 56.21 | 66.12 | 70.41 |

It can be observed that when pseudo-labels from the DL backbone are not used, the performance drops significantly. In this case, the model relies solely on the shapelet model, which performs well in data-scarce conditions (early stage) but shows lower performance once sufficient data become available (later stages).

## F.2  SHAPELETS SELECTED FROM DATASET VS. RANDOMLY INITIALISED SHAPELETS

The benefit of using shapelets discovered from the training set has been widely demonstrated in prior works Le et al. (2024; 2022). These studies show that shapelets extracted from important regions of the time series provide a better starting point, and with only minimal fine-tuning can significantly outperform randomly initialised shapelets. Moreover, selecting shapelets from the training data helps the model focus on the most informative regions of the time series. When combined with a left–right window search strategy, this greatly reduces computation compared to using randomly initialised shapelets, which require evaluation over the entire time series.

To empirically validate this, we compare shapelets selected from the training dataset against randomly initialized shapelets on the APAVA and FaceDetection datasets (Table 16).

Table 16: Performance comparison between selected and randomly initialized shapelets on APAVA and FaceDetection datasets.

| Dataset | Initialization Type | 1% | 5% | 20% |
|---|---|---|---|---|
| APAVA | Selected Shapelet Initialization | **60.26** | **69.12** | **74.11** |
| | Random Shapelet Initialization | 58.24 | 67.71 | 72.94 |
| FaceDetection | Selected Shapelet Initialization | **47.35** | **55.13** | **58.29** |
| | Random Shapelet Initialization | 45.24 | 53.59 | 57.03 |

## F.3  MULTIVARIATE (OVER ALL CHANNELS) SHAPELETS

We found that multivariate (over-all-channels) shapelets are learned jointly from all channels instead of being restricted to each channel independently. By doing so, the extracted shapelets are able to capture patterns that involve interactions between different channels, rather than modeling them in isolation. This joint representation allows the model to effectively exploit inter-channel dependencies, leading to a richer and more discriminative representation of the time series data.

To further support this claim, we conducted an ablation study (Table 17) to isolate the effect of using multivariate shapelets. This comparison clearly illustrates the performance difference when the model is equipped with multivariate shapelets versus when it is not.

Table 17: Ablation study with and without multivariate shapelets on APAVA and FaceDetection datasets.

| Dataset | Method | 1% | 5% | 20% |
|---|---|---|---|---|
| APAVA | ShapeMatch | **60.26** | **69.12** | **74.11** |
| | – Multivariate Shapelet | 58.21 | 68.21 | 73.71 |
| FaceDetection | ShapeMatch | **60.26** | **69.12** | **74.11** |
| | – Multivariate Shapelet | 58.21 | 68.21 | 73.71 |

### F.4 EFFECT OF OFFLINE SHAPELET DISCOVERY

Our framework relies on an offline shapelet discovery stage to provide a strong initialization for the shapelets before end-to-end training with ShapeMatch. Instead of starting from random patterns, the discovered shapelets are selected to be discriminative with respect to the target classes and to cover diverse temporal structures in the data. This warm-start helps the subsequent optimization avoid poor local minima and allows the shapelets to focus on refining meaningful patterns rather than first searching for them from scratch.

To assess the importance of this discovery stage, we compare our standard pipeline (ShapeMatch + Shapelet Discovery) with a variant where all shapelets are randomly initialized and trained jointly with the rest of the model. As shown in Table 18, using offline discovery consistently improves performance on both APAVA and FaceDetection across all label ratios. The gains range from roughly 1.3 to 2.0 percentage points, with the largest improvements observed in the 1%–5% label regime. These results confirm that the proposed discovery step is not merely a convenience, but a crucial component that stabilizes training and yields more accurate semi-supervised models.

Table 18: Ablation study comparing ShapeMatch with offline shapelet discovery versus random shapelet initialization on APAVA and FaceDetection at different label ratios.

| Dataset | APAVA (76.30%) | | | FaceDetection (63.25%) | | |
|---|---|---|---|---|---|---|
| Label Ratio | 1% | 5% | 20% | 1% | 5% | 20% |
| ShapeMatch + Shapelet Discovery | **60.26** | **69.12** | **74.11** | **47.35** | **55.13** | **58.29** |
| ShapeMatch + Random Initialization | 58.41 | 67.82 | 72.61 | 45.51 | 53.15 | 56.62 |

## G OTHER RESULTS

### G.1 COMPARISON WITH PREVIOUS SHAPELET-BASED MODELS

We summarise the key differences between our method and prior shapelet approaches in Table 19.

Table 19: Comparison of ShapeMatch with prior shapelet-based models.

| | Task | Type of Network | Type of Shapelet |
|---|---|---|---|
| ShapeFormer | Supervised Learning (Classification) | Classifier Backbone | Univariate Shapelet |
| ShapeNet (Random Init) | Supervised Learning (Classification) | Classifier Backbone | Univariate Shapelet |
| Unsupervised Shapelet | Unsupervised Learning (Clustering) | Clustering Backbone | Univariate Shapelet |
| **Our ShapeMatch** | Semi-supervised Learning (Label-scarce Classification) | Framework applied to any Classifier Backbone | Univariate + Multivariate Shapelet |

Our method differs from the above approaches in several important aspects:

- **Target Task:** ShapeMatch is designed for semi-supervised learning, enabling effective training with limited labeled data, whereas the other methods focus solely on supervised or unsupervised tasks.

- **Model Type:** ShapeMatch is a framework that can enhance any time-series classification backbone. For example, when applied to the ShapeFormer backbone, our approach delivers substantial accuracy gains, as shown in Table 20.

Table 20: Performance comparison under different label ratios using ShapeFormer backbone Le et al. (2024).

| Label Ratio | 1% | 5% | 20% |
|---|---|---|---|
| Supervised Learning (ShapeFormer Backbone) | 58.26 | 69.52 | 72.41 |
| ShapeMatch (ShapeFormer Backbone) | **65.35** | **74.48** | **75.92** |

- **Shapelet:** Previous work only used univariate shapelets (representing a single channel), while our method leverages both univariate and multivariate shapelets, enabling the capture of dependencies across multiple channels.

## G.2 LLM-BASED MODELS AS BACKBONE

LLM-based unified models offer an interesting approach for time series tasks. However, current experiments with models like GPT4TS Zhou et al. (2023b) primarily focus on unsupervised (zero-shot) and semi-supervised (few-shot) forecasting tasks, rather than classification. To clarify the benefits of our method, we conducted additional experiments comparing our approach with GPT4TS, as shown in Table 21. *Note:* Since GPT4TS uses additional data from other fields, the comparison may not be entirely fair.

Table 21: Comparison with GPT4TS under varying label ratios for classification.

| Label Ratio | 1% | 5% | 20% |
|---|---|---|---|
| Supervised Learning (GPT4TS backbone Zhou et al. (2023b)) | 51.41 | 63.43 | 66.75 |
| ShapeMatch (GPT4TS backbone Zhou et al. (2023b)) | **56.42** | **66.47** | **72.23** |

## G.3 CNN-BASED MODELS AS BACKBONE

Table 22 presents the performance of various SSL methods using TSLANet Eldele et al. (2024) as the backbone for different label ratios (1%, 5%, and 20%). As the proportion of labelled data increases, all methods show improved accuracy. Notably, ShapeMatch consistently outperforms the others across all label ratios, demonstrating its strong adaptability and effectiveness when integrated with any deep learning model for multivariate time series classification (MTSC).

Table 22: Performance comparison under different label ratios when use TSLANet as backbone

| Label Ratio | 1% | 5% | 20% |
|---|---|---|---|
| Supervised Learning (TSLANet backbone) | 52.16 | 62.82 | 66.64 |
| Pseudo-Label (TSLANet backbone) Lee et al. (2013) | 54.41 | 64.71 | 69.08 |
| FixMatch (TSLANet backbone) Sohn et al. (2020) | 54.18 | 65.26 | 69.02 |
| Semiformer (TSLANet backbone) Weng et al. (2022) | 55.58 | 65.77 | 69.91 |
| ShapeMatch (TSLANet backbone) | **57.09** | **66.41** | **71.49** |

## H  ERROR BAR

We report the standard deviation of ShapeMatch performance for five runs with different random seeds in Table 23, which shows that the performance of ShapeMatch is stable.

Table 23: Error bar for ShapeMatch over 5 runs.

| | APAVA (76.30%) | | | TDBrain (87.17%) | | |
|---|---|---|---|---|---|---|
| Supervised Learning | $52.41 \pm 0.23$ | $63.52 \pm 0.17$ | $71.14 \pm 0.14$ | $60.21 \pm 0.19$ | $68.41 \pm 0.22$ | $78.25 \pm 0.18$ |
| Pseudo-Label Lee et al. (2013) | $54.12 \pm 0.20$ | $65.13 \pm 0.16$ | $72.19 \pm 0.21$ | $61.86 \pm 0.27$ | $69.71 \pm 0.13$ | $79.91 \pm 0.15$ |
| FixMatch Sohn et al. (2020) | $55.44 \pm 0.14$ | $66.52 \pm 0.24$ | $72.42 \pm 0.19$ | $63.16 \pm 0.26$ | $71.11 \pm 0.12$ | $80.56 \pm 0.29$ |
| Semiformer Weng et al. (2022) | $55.31 \pm 0.25$ | $65.92 \pm 0.22$ | $72.65 \pm 0.13$ | $62.21 \pm 0.18$ | $72.72 \pm 0.21$ | $81.98 \pm 0.14$ |
| ShapeMatch | $\mathbf{60.26 \pm 0.11}$ | $\mathbf{69.12 \pm 0.20}$ | $\mathbf{74.11 \pm 0.19}$ | $\mathbf{67.72 \pm 0.13}$ | $\mathbf{77.24 \pm 0.27}$ | $\mathbf{84.14 \pm 0.16}$ |

## I  SHAPEMATCH ON IMBALANCED DATA SETTING

Since the nature of shapelets is as class-specific features, their discovery is not prone to the imbalanced data problem. Therefore, the results of shapelet-guidance in general, and our ShapeMatch in particular, significantly outperform other methods under imbalanced data settings.

To demonstrate this, we created imbalanced settings as long-tail distributions for the PTB-XL (5 classes) and SpokenArabicDigits (10 classes) datasets. We use the following formula to construct the imbalanced long-tail dataset:

$$\gamma = \frac{N_{\max}}{N_{\min}}$$

where $N_{\max}$ and $N_{\min}$ are the numbers of training samples for the largest and smallest classes, respectively, and $\gamma$ is the parameter controlling the skewness.

In this experiment, we fix the label ratio at 5% and evaluate the performance across various values of $\gamma$.

The results demonstrate that our method significantly outperforms all other methods across all imbalanced settings. This highlights the advantages of the shapelet-guided approach, especially under severe class imbalance.

Table 24: Performance comparison under imbalanced settings (various $\gamma$ values). Label ratio is fixed at 5%.

| Method | $\gamma = 1$ (Balanced) | | | $\gamma = 10$ | | | $\gamma = 50$ | | | $\gamma = 100$ | | |
|---|---|---|---|---|---|---|---|---|---|---|---|---|
| | Accuracy | Recall | F1 | Accuracy | Recall | F1 | Accuracy | Recall | F1 | Accuracy | Recall | F1 |
| Supervised | 60.97 | 56.28 | 56.87 | 55.71 | 44.67 | 51.31 | 44.71 | 34.16 | 40.41 | 42.22 | 31.19 | 38.11 |
| Pseudo-Label | 61.93 | 57.12 | 58.04 | 57.23 | 47.37 | 53.71 | 48.27 | 39.67 | 44.64 | 44.71 | 35.71 | 40.86 |
| FixMatch | 63.36 | 59.16 | 62.84 | 58.64 | 51.33 | 55.12 | 49.56 | 41.22 | 45.91 | 45.43 | 37.18 | 41.63 |
| Semiformer | 63.83 | 59.22 | 62.12 | 58.83 | 50.21 | 55.34 | 49.82 | 42.22 | 46.15 | 46.64 | 38.48 | 43.12 |
| **ShapeMatch** | **65.89** | **64.85** | **65.51** | **63.01** | **58.98** | **61.64** | **57.14** | **53.38** | **55.67** | **55.12** | **50.37** | **53.69** |

## J  LIMITATION AND FUTURE WORK

A current challenge in our work is that shapelet discovery primarily relies on CPU-based methods, which can be computationally intensive and occasionally time-consuming. Although we introduce strategies to reduce the run time, the process still benefits most from substantial computational resources to achieve acceleration. In future work, we aim to further enhance the efficiency of shapelet discovery and also explore alternative strategies that can provide simpler yet effective approaches for shapelet generation.

## K  THE USE OF LARGE LANGUAGE MODELS

We used a large language model (ChatGPT) to help with editing this paper. It was only used for simple tasks such as fixing typos, rephrasing sentences for clarity, and improving word choice. All ideas, experiments, and analyses were done by the authors, and the use of LLMs does not affect the reproducibility of our work.

