# OpenReview forum: "ShapeMatch: Shapelet-Guided Semi-Supervised Learning for Multivariate Time Series Classification"
_ICLR.cc/2026/Conference — Submitted to ICLR 2026_

### Official Review · Reviewer_a8G6 · 2025-10-30

**Soundness:** 3
**Presentation:** 3
**Contribution:** 2
**Rating:** 4
**Confidence:** 5

**Summary:**

This paper introduces ShapeMatch, a shapelet-based framework designed for multivariate semi-supervised time series classification. ShapeMatch incorporates a discriminative shapelet model alongside a shapelet-guided masking augmentation strategy to effectively leverage unlabeled data in semi-supervised learning settings. Experimental evaluations conducted on five healthcare and seven UEA benchmark time series datasets demonstrate that ShapeMatch consistently outperforms four traditional semi-supervised classification baselines and exhibits good model-agnostic adaptability across different backbone architectures.

**Strengths:**

1. The paper presents an interesting application of the shapelet concept to multivariate semi-supervised time series classification, with a clear and coherent overall structure.
2. The figures and experimental tables are well-organized and easy to interpret, and the paper provides a comprehensive review of prior work on fully supervised multivariate time series classification.
3. The experimental evaluation across different backbone architectures is thorough, offering valuable insights for future researchers in selecting state-of-the-art deep learning models for fully supervised classification tasks.

**Weaknesses:**

1. Figure 1, intended as the motivation figure, is unconvincing. The comparison between ShapeMatch (using both labeled and unlabeled data) and models (Transformer and a shapelet-only model) trained solely on limited labeled data is unfair. The paper should instead compare ShapeMatch with existing semi-supervised classification methods to demonstrate its true advantage. Moreover, at a 100% labeling rate, ShapeMatch performs similarly to the Transformer, suggesting that shapelets offer limited benefit without unlabeled data.
2. The paper’s novelty is incremental. ShapeMatch adopts the ShapeletDistance strategy from Le et al. (2022) for shapelet search. While the proposed shapelet-guided masking (Eq. 3) is commendable, the augmentation methods (jittering, masking, cropping, shifting) are well-established and do not warrant detailed discussion.
3. The shapelet discovery process is time-consuming. As shown in Table 8, searching with all training samples requires 10.5 hours, compared to only 0.72 hours for supervised training (Table 6), limiting real-world applicability. Although the authors acknowledge this in the appendix, prior works have already proposed learnable shapelet methods [1,2] to significantly reduce search time.
4. The paper lacks a review of shapelet-based methods, particularly those employing learnable shapelet approaches [1,2], which substantially reduce the computational burden compared to the search-based approach described in Equation 1.
5. The experimental baselines include only four semi-supervised methods, none of which are time-series–specific. Moreover, related works [3,4] have explored shapelet-based semi-supervised time series classification but are not discussed.
6. The anonymous code repository is inaccessible, raising concerns about reproducibility.

[1] Grabocka, Josif, et al. "Learning time-series shapelets." Proceedings of the 20th ACM SIGKDD international conference on Knowledge discovery and data mining. 2014.

[2] Qu, Eric, et al. "CNN kernels can be the best shapelets." The Twelfth International Conference on Learning Representations. 2024.

[3] Du, Mingsen, et al. "Multivariate Time Series Classification via Heterogeneous Graph Representation." IEEE Transactions on Industrial Informatics (2025).

[4] Wang, Zhicheng, et al. "Multiview Contrastive Shapelet Learning: A Novel Semisupervised Approach for Explainable Machine Fault Diagnosis With Insufficient Annotated Data." IEEE Transactions on Instrumentation and Measurement (2025).

**Questions:**

1. The authors selected seven UEA time series datasets for semi-supervised evaluation. Since UEA datasets provide only default training and test splits, how was the testing procedure for ShapeMatch conducted?
2. In Table 3, were the reported results obtained from a single dataset? If so, which dataset was used?
3. As noted in [1,2], selecting an appropriate subsequence length is crucial for shapelet discovery. Based on Table 9, how were the shapelet lengths determined for the five healthcare and seven UEA datasets?
4. Since shapelets are designed to enhance interpretability through discriminative subsequences, does the paper provide visualizations showing any notable patterns in the learned shapelets, or any quantitative metrics assessing their quality?

---

> ### Author Response · Authors · 2025-11-27
>
> > **W1:** Figure 1, intended as the motivation figure, is unconvincing. The comparison between ShapeMatch (using both labeled and unlabeled data) and models (Transformer and a shapelet-only model) trained solely on limited labeled data is unfair. The paper should instead compare ShapeMatch with existing semi-supervised classification methods to demonstrate its true advantage.
>
> #### **AW1:** We agree that Figure 1 should be better aligned with the semi-supervised setting. In the revised version, we include a Transformer trained with FixMatch in the motivation figure, comparing it to ShapeMatch under the same labeled–unlabeled splits. As shown, the FixMatch Transformer only slightly improves over its supervised counterpart, whereas both the shapelet-only model and ShapeMatch achieve substantially higher accuracy in the label-scarce regime.
>
> >  Moreover, at a 100% labeling rate, ShapeMatch performs similarly to the Transformer, suggesting that shapelets offer limited benefit without unlabeled data.
>
> #### **At a 100% labeling rate, the problem becomes fully supervised**, with no unlabeled data available, so it is no longer a semi-supervised setting. In this case, semi-supervised methods such as FixMatch and our ShapeMatch are expected to match the standard Transformer, as there is no additional unlabeled information to exploit.
>
> ---
>
> > **W2:** The paper’s novelty is incremental. ShapeMatch adopts the ShapeletDistance strategy from Le et al. (2022) for shapelet search. While the proposed shapelet-guided masking (Eq. 3) is commendable, the augmentation methods (jittering, masking, cropping, shifting) are well-established and do not warrant detailed discussion.
>
>
> #### **AW2:** We agree that ShapeletDistance is a well-established method, and we use it as an off-the-shelf module, similar to a linear or convolution layer. **Our main contribution lies in the novel framework** that leverages shapelets to improve arbitrary deep backbones in the semi-supervised setting. Specifically:
>
> #### - We are, to the best of our knowledge, the first to use shapelet model predictions to guide a deep backbone during the early training stage.
> #### - We are the first to use shapelet distances to refine and optimize pseudo-labels.
> #### - We also introduce a shapelet-based augmentation strategy that is tailored to time series and empirically shown to be highly effective, beyond standard jittering/masking/cropping/shifting.
>
> ---
>
> > **W3:** The shapelet discovery process is time-consuming. As shown in Table 8, searching with all training samples requires 10.5 hours, compared to only 0.72 hours for supervised training (Table 6), limiting real-world applicability.
>
> #### **AW3:** We respectfully disagree that the shapelet discovery process is prohibitively time-consuming. The 10.5 hours reported in Table 8 correspond to a naive search over all training samples used only for analysis. In practice, as shown in Table 7, our improved discovery procedure finishes in under one minute at a cost of only $0.368, without any performance drop. Furthermore, at inference time the shapelet model is removed, so the inference cost is identical to that of the standard backbone.
>
> > Although the authors acknowledge this in the appendix, prior works have already proposed learnable shapelet methods [1,2] to significantly reduce search time.
>
> #### In fact, our shapelets are also learnable. The key difference between our approach and the deep learning–based shapelet methods you mentioned is that we include an offline phase to discover the initial shapelets, which are then further optimized during the training process. This two-stage strategy has been shown to yield better performance in several recent works [1,2].
>
> #### To quantify its effect, we compare ShapeMatch with and without the discovery step. Replacing discovery by random initialization consistently leads to 1.3–2.0% accuracy drops across both datasets and all label ratios (Table below), showing that the discovery phase is crucial rather than cosmetic. Further details are provided in Section F.4 of the revised appendix.
>
> | Dataset                               | APAVA (76.30%)  |            |            | FaceDetection (63.25%) |            |            |
> |---------------------------------------|------------------|------------|------------|-------------------------|------------|------------|
> | **Label Ratio**                       | **1%**           | **5%**     | **20%**    | **1%**                  | **5%**     | **20%**    |
> | **ShapeMatch + Shapelet Discovery**   | **60.26**        | **69.12**  | **74.11**  | **47.35**               | **55.13**  | **58.29**  |
> | ShapeMatch + Random Initialization    | 58.41            | 67.82      | 72.61      | 45.51                   | 53.15      | 56.62      |

---

> ### Author Response · Authors · 2025-11-27
>
> > **W4:** The paper lacks a review of shapelet-based methods, particularly those employing learnable shapelet approaches [1,2], which substantially reduce the computational burden compared to the search-based approach described in Equation 1.
>
> #### **AW4:** We thank the reviewer for this suggestion. In the revised version, **we added a dedicated subsection in the related-work part to discuss shapelet-based methods**, including learnable shapelet approaches [1,2], and clarify how they differ from our setting. As mentioned in AW3, we use the the initial (already short) shapelet discovery step to improve a performance by 2–3%.
>
>
>
> ---
>
> > **W5:** The experimental baselines include only four semi-supervised methods, none of which are time-series–specific. Moreover, related works [3,4] have explored shapelet-based semi-supervised time series classification but are not discussed.
>
> #### **AW5:** Following your suggestion, in the revised paper we have additionally included two self-supervised methods, TS2VEC and CA-TCC (corresponding to [4]), and one semi-supervised model, semiHGR (corresponding to [3]), to further clarify the benefits of our approach. As shown in **Tables 1 and 2 of the revised paper**, ShapeMatch consistently outperforms all these baselines across all comparison settings. We attribute these gains to the shapelet guidance applied in the early training stage, which helps the backbone network learn more effective representations from the outset, thereby substantiating our contributions.
>
> ---
>
> > **W6:** The anonymous code repository is inaccessible, raising concerns about reproducibility.
>
> #### **AW6:** We apologize for the inconvenience. The issue has been fixed, and the anonymous code repository is now accessible.
>
> ---
>
> > **Q1:** The authors selected seven UEA time series datasets for semi-supervised evaluation. Since UEA datasets provide only default training and test splits, how was the testing procedure for ShapeMatch conducted?
>
> #### **AQ1:** We split only the training set into labeled and unlabeled subsets according to the specified label ratio, while in the testing phase the model is evaluated on the entire test set.
>
> ---
>
> > **Q2:** In Table 3, were the reported results obtained from a single dataset? If so, which dataset was used?
>
> #### **AQ2:** Thank you for pointing this out. We used the APAVA dataset for all experiments in Table 3 and will revise the paper to clarify this.
>
> ---
>
> > **Q3:** As noted in [1,2], selecting an appropriate subsequence length is crucial for shapelet discovery. Based on Table 9, how were the shapelet lengths determined for the five healthcare and seven UEA datasets?
>
> #### **AQ3:** A key advantage of our shapelet discovery procedure over [1,2] is that it does not require predefining the shapelet lengths. In the discovery step, we first use perceptually important points to identify informative subsequences (shapelet candidates) of arbitrary length. Among these candidates, we then select those with the highest information gain. Thus, the lengths are automatically induced by this process rather than manually specified or tuned per dataset.
>
> ---
>
> > **Q4:** Since shapelets are designed to enhance interpretability through discriminative subsequences, does the paper provide visualizations showing any notable patterns in the learned shapelets, or any quantitative metrics assessing their quality?
>
> #### **AQ4:** The main goal of our work is to use shapelets to improve the performance of arbitrary deep backbones. The final predictions are therefore made by the deep backbone, not by the shapelet model itself, and we do not claim that our method directly enhances interpretability. However, **Figure 3** includes visualizations illustrating how the discovered shapelets help guide time-series augmentation.

---

### Official Review · Reviewer_FMa2 · 2025-11-01

**Soundness:** 2
**Presentation:** 4
**Contribution:** 2
**Rating:** 4
**Confidence:** 5

**Summary:**

This paper proposes a shapelet-guided semi-supervised framework, termed ShapeMatch, for multivariate time series classification. The framework aims to guide deep learning models in aligning their predictions with those of a shapelet-based model during the initial training phase. Specifically, ShapeMatch introduces a shape mask augmentation strategy designed for multivariate time series, enabling the model to extract more informative representations from unlabeled data in a semi-supervised setting. Experimental evaluations conducted on 12 multivariate time series datasets demonstrate that ShapeMatch consistently outperforms selected baseline methods.

**Strengths:**

1.	The paper is clearly structured, and the figures and tables are well-organized, which facilitates readers’ understanding of the content.
2.	The study’s introduction of time series shapelets into the problem of semi-supervised multivariate time series classification represents a valuable approach, providing a useful reference for future research on shapelet-based time series classification.

**Weaknesses:**

1.	The paper offers limited novelty in time series shapelets modeling and overlooks much prior work. Specifically, the authors adopt the ShapeletDistance method (Equation 1) for shapelet search, which has been established in prior studies. Furthermore, the runtime analysis in the appendix indicates that this procedure is computationally expensive. In contrast, studies [1,2,3] have demonstrated that learning shapelets via neural networks can significantly reduce the time required to discover shapelets compared to distance-based search methods (Equation 1).

2.	The study lacks innovation in modeling relationships between variables, which is critical for multivariate time series semi-supervised classification. While the authors note in the related work that existing semi-supervised time series methods do not consider inter-variable relationships, their model only mentions in lines 194–197 that the input is multivariate, and the experimental analysis lacks discussion on inter-variable relationship modeling. In comparison, studies [4] and [5] employ clustering and graph networks, respectively, to capture relationships among variables.

3.	The semi-supervised classification baselines selected in this work are primarily designed for image data and do not account for methods specifically developed for time series. For example, studies [6,7] apply shapelets to semi-supervised classification of multivariate time series, while studies [8,9,10] focus on semi-supervised classification of univariate time series, but their semi-supervised learning paradigms can also be effectively applied to multivariate time series.

[1] Learning time-series shapelets. KDD, 2014.

[2] Shapenet: A shapelet-neural network approach for multivariate time series classification. AAAI, 2021.

[3] Multiview unsupervised shapelet learning for multivariate time series clustering. TPAMI, 2022.

[4] From similarity to superiority: Channel clustering for time series forecasting. NeurIPS, 2024.

[5] Fully-Connected Spatial-Temporal Graph for Multivariate Time Series Data. AAAI, 2024.

[6] Heterogeneous Relationships of Subjects and Shapelets for Semi-supervised Multivariate Series Classification. arXiv, 2024.

[7] Multiview Contrastive Shapelet Learning: A Novel Semisupervised Approach for Explainable Machine Fault Diagnosis With Insufficient Annotated Data. IEEE Transactions on Instrumentation and Measurement, 2025.

[8] Self-supervised learning for semi-supervised time series classification. PAKDD, 2020.

[9] Semi-supervised time series classification by temporal relation prediction. ICASSP, 2021.

[10] Self-supervised contrastive representation learning for semi-supervised time-series classification. TPAMI, 2023.

**Questions:**

1.	In the context of using shapelets for semi-supervised multivariate time series classification, what are the core differences between the proposed method and studies [6,7]? Additionally, regarding the runtime for shapelet discovery, what are the relative advantages and disadvantages of the proposed approach compared to [6,7]?

2.	Compared to studies [8,9,10], what specific advantages does the proposed method demonstrate in semi-supervised classification performance across the selected 11 multivariate time series datasets?

3.	Studies [9,10] also employ data augmentation techniques discussed in their proposed method (e.g., jittering) for semi-supervised time series classification. Without the shapelet-guided mask, how does the proposed ShapeAug augmentation differ from these existing methods?

**Details Of Ethics Concerns:**

None.

---

> ### Author Response · Authors · 2025-11-27
>
> > **W1:** The paper offers limited novelty in time series shapelets modeling and overlooks much prior work. Specifically, the authors adopt the ShapeletDistance method (Equation 1) for shapelet search, which has been established in prior studies. Furthermore, the runtime analysis in the appendix indicates that this procedure is computationally expensive. In contrast, studies [1,2,3] have demonstrated that learning shapelets via neural networks can significantly reduce the time required to discover shapelets compared to distance-based search methods (Equation 1).
>
> #### **AW1:** Thank you for your questions. Thank you for your question. In fact, our shapelets are also learnable. The key difference between our approach and the deep learning–based shapelet methods you mentioned is that we incorporate an offline phase to discover initial shapelets, which are then further optimized during training. This two-stage strategy has been shown to yield better performance in several recent works [1,2]. Moreover, if we remove the initial discovery step and instead randomly initialize the shapelets, we observe a performance drop of about 2–3%. Finally, our shapelet formulation eliminates the need to manually design the shapelet length, which is one of the most challenging hyperparameters to tune in [1,2].
>
> #### To quantify its effect, we compare ShapeMatch with and without the discovery step. Replacing discovery by random initialization consistently leads to 1.3–2.0% accuracy drops across both datasets and all label ratios (Table below), showing that the discovery phase is crucial rather than cosmetic. Further details are provided in Section F.4 of the revised appendix.
>
> | Dataset                               | APAVA (76.30%)  |            |            | FaceDetection (63.25%) |            |            |
> |---------------------------------------|------------------|------------|------------|-------------------------|------------|------------|
> | **Label Ratio**                       | **1%**           | **5%**     | **20%**    | **1%**                  | **5%**     | **20%**    |
> | **ShapeMatch + Shapelet Discovery**   | **60.26**        | **69.12**  | **74.11**  | **47.35**               | **55.13**  | **58.29**  |
> | ShapeMatch + Random Initialization    | 58.41            | 67.82      | 72.61      | 45.51                   | 53.15      | 56.62      |
>
> ---
> > **W2:** The study lacks innovation in modeling relationships between variables, which is critical for multivariate time series semi-supervised classification. While the authors note in the related work that existing semi-supervised time series methods do not consider inter-variable relationships, their model only mentions in lines 194–197 that the input is multivariate, and the experimental analysis lacks discussion on inter-variable relationship modeling. In comparison, studies [4] and [5] employ clustering and graph networks, respectively, to capture relationships among variables.
>
> #### **AW2:** We thank the reviewer for this question. In the related work, we note that existing univariate semi-supervised methods do not capture inter-variable relationships in multivariate time series. Works [4] and [5] are fully supervised: [4] focuses on **forecasting** and is outside our scope, while [5] can be seen as **another backbone model** (similar to Shapeformer, Medformer, iTransformer). **Our framework is intended to improve such backbones in the semi-supervised setting, rather than act as a direct counterpart.**
>
> #### **Inter-variable relationships:** Existing multivariate backbones (e.g., Shapeformer, Medformer, iTransformer, and [5]) already include mechanisms for modeling inter-variable dependencies, and ShapeMatch is designed to enhance them while preserving these properties. Additionally, we jointly learn univariate and multivariate shapelets in the same offline discovery phase: univariate shapelets operate on single channels, while multivariate shapelets span the same temporal window across all channels with an aggregated distance. This allows univariate shapelets to capture channel-specific patterns and multivariate shapelets to explicitly encode inter-variable relationships.

---

> ### Author Response · Authors · 2025-11-27
>
> > **W3:** The semi-supervised classification baselines selected in this work are primarily designed for image data and do not account for methods specifically developed for time series.
>
> #### **AW3:** Thank you for your detailed comments. **We were unable to access the paper in [6], as it appears to have been withdrawn from arXiv**. In contrast, [7] proposes a highly task-specific method designed solely for machine fault diagnosis, and we were unable to re-implement it due to the lack of publicly available code. Moreover, the model in [7] is a semi-supervised–specific architecture, meaning it cannot be readily used to improve other backbones such as CNN-, Transformer-, or LLM-based models, unlike our ShapeMatch framework.
>
> #### Following your suggestion, in the revised paper we have additionally included two self-supervised methods, TS2VEC and CA-TCC (corresponding to [8] and [10]), and further one more semi-supervised model, semiHGR [A1], to further clarify the benefits of our approach. As shown in **Tables 1 and 2 of the revised paper**, ShapeMatch consistently outperforms all these baselines across all comparison settings. We attribute these gains to the shapelet guidance applied in the early training stage, which helps the backbone network learn more effective representations from the outset, thereby substantiating our contributions.
>
> [A1] Du, Mingsen, et al. "Multivariate Time Series Classification via Heterogeneous Graph Representation." IEEE Transactions on Industrial Informatics (2025).
>
> ---
>
> > **Q1:** In the context of using shapelets for semi-supervised multivariate time series classification, what are the core differences between the proposed method and studies [6,7]? Additionally, regarding the runtime for shapelet discovery, what are the relative advantages and disadvantages of the proposed approach compared to [6,7]?
>
> #### **AQ1:** Thanks for pointing this out. **Regarding [6], we note that it is an early, withdrawn arXiv preprint and remains unpublished.** Compared to [7], the main difference is that [7] uses shapelets as the core modeling component and introduces several techniques tailored to a single target dataset. In contrast, our work proposes a general framework in which shapelets serve as a supporting module to guide the training of arbitrary deep learning backbones, including Transformers, CNNs, and even LLM-style architectures. Another key distinction is our shapelet-based augmentation; to the best of our knowledge, we are the first to introduce this type of augmentation for time series.
>
> ---
>
> > **Q2:** Compared to studies [8,9,10], what specific advantages does the proposed method demonstrate in semi-supervised classification performance across the selected 11 multivariate time series datasets?
>
> #### **AQ2:** Compared to self-supervised methods in [8,9,10], our work differs in several key aspects. First, we primarily focus on multivariate time series and explicitly exploit channel correlations via multivariate shapelets, which provide richer structural information than univariate approaches. Second, as highlighted in our motivation, we observe that standard deep learning backbones often struggle in label-scarce settings; consequently, directly applying existing self-supervised methods on these backbones does not yield satisfactory performance. **This motivates our proposal of shapelet-guided training to support and stabilize deep models in the early stages of learning.**
>
> #### To validate this, we conducted additional experiments with CA-TCC [10] on 11 multivariate time-series datasets. Due to time constraints and limit implementation material, we were unable to implement [8,9]. The results show that our method substantially outperforms CA-TCC across all compared settings, further demonstrating the effectiveness and contributions of our approach.
>
> ---
> > **Q3:** Studies [9,10] also employ data augmentation techniques discussed in their proposed method (e.g., jittering) for semi-supervised time series classification. Without the shapelet-guided mask, how does the proposed ShapeAug augmentation differ from these existing methods?
>
> #### **AQ3:** The key difference is that ShapeAug is **shapelet-aware**. Prior works [9,10] typically apply augmentations (e.g., jitter, scaling, warping) uniformly or globally over the entire series, without distinguishing which segments are most discriminative. In ShapeAug, we first use the discovered shapelets to build a **binary/soft mask over key subsequences**, then: (i) apply **weaker, label-preserving transforms** on the key segments to avoid destroying critical local patterns, and (ii) apply **stronger perturbations outside** these key regions to encourage robustness. Thus, ShapeAug is not just a collection of standard augmentations, but a mechanism that **modulates augmentation strength based on shapelet-informed importance**, which we find is crucial for stable performance under label scarcity.

---

### Official Review · Reviewer_JtMc · 2025-11-01

**Soundness:** 3
**Presentation:** 3
**Contribution:** 3
**Rating:** 6
**Confidence:** 4

**Summary:**

This paper tackles the problem of Multivariate Time Series Classification under label-scarce conditions. The authors propose ShapeMatch, a semi-supervised framework designed to enhance model generalization with limited labeled data. The framework combines two main ideas: (1) a hybrid training paradigm that integrates classical Shapelet models to guide deep learning backbones during early training stages, and (2) a tailored ShapeAug strategy that introduces meaningful temporal variations while preserving key structural patterns. Experiments on multiple benchmark datasets show that ShapeMatch outperforms existing state-of-the-art semi-supervised and fully supervised baselines, especially in low-label regimes.

**Strengths:**

1. The paper addresses a practically relevant and underexplored problem and offers clear potential for real-world applications.

2. The framework and model pipeline are clearly illustrated, improving readability and conceptual understanding.

**Weaknesses:**

1. Some claims are overstated; most shapelet-based methods were originally developed for univariate time series and may not generalize directly to multivariate contexts.

2. The experimental presentation lacks clarity as key results are fragmented, and the overall comparison could be made more intuitive and systematic.

**Questions:**

1. Could the authors provide a concise summary (e.g., a table) showing ShapeMatch’s performance across different deep learning backbones, to clearly support its claimed generality?

2. Given that only r = 50 samples per class are used, how stable are the reported results? Including standard deviations or comparisons with larger r (e.g., 100) would clarify robustness.

3. How are multivariate shapelets learned and integrated with univariate ones? A more detailed explanation would help clarify the modeling of inter-variable dependencies.

4. Is there theoretical or empirical evidence that shapelet priors reliably improve robustness under label scarcity, especially in high-dimensional MTSC settings?

5. Since the framework shares conceptual similarities with FixMatch, could the authors elaborate on the key methodological differences and provide a direct comparison to emphasize ShapeMatch’s unique contributions?

---

> ### Author Response · Authors · 2025-11-27
>
> > **Q1:** Could the authors provide a concise summary (e.g., a table) showing ShapeMatch’s performance across different deep learning backbones, to clearly support its claimed generality?
>
> #### **AQ1:** Yes, we have already conducted this analysis in Table 3 using different backbones, including iTransformer, ShapeFormer, MedFormer, and PatchTST.
>
> ---
>
> > **Q2:** Given that only \(r = 50\) samples per class are used, how stable are the reported results? Including standard deviations or comparisons with larger \(r\) (e.g., 100) would clarify robustness.
>
> #### **AQ2:** We agree that robustness is important. In fact, we already provide an error-bar comparison in **Section H of the appendix**. Each result is averaged over three runs, and the corresponding error bars are relatively narrow, indicating that the reported gains are stable with respect to the specific subset of labeled data.
>
> ---
>
> > **Q3:** How are multivariate shapelets learned and integrated with univariate ones? A more detailed explanation would help clarify the modeling of inter-variable dependencies.
>
> #### **AQ3:** We thank the reviewer for this question. In ShapeMatch, univariate and multivariate shapelets are learned jointly in the same offline discovery phase: univariate shapelets operate on single channels, while multivariate shapelets span the same temporal window across all channels and use an aggregated (channel-wise) distance. After discovery, distances to all shapelets are concatenated into a single feature vector, on which the shapelet classifier, shapelet-guided consistency loss, and ShapeAug all operate. In this way, univariate shapelets capture strong channel-specific patterns, whereas multivariate shapelets explicitly model inter-variable dependencies.
>
> ---
>
> > **Q4:** Is there theoretical or empirical evidence that shapelet priors reliably improve robustness under label scarcity, especially in high-dimensional MTSC settings?
>
> #### **AQ4:** Our evidence is primarily empirical. Across multiple MTSC benchmarks and low-label regimes (1%, 5%, 20%), ShapeMatch consistently outperforms both supervised and SSL baselines, indicating that the shapelet prior helps stabilize semi-supervised learning. Additional robustness tests are provided in Appendix Section H.
>
> ---
>
> > **Q5:** Since the framework shares conceptual similarities with FixMatch, could the authors elaborate on the key methodological differences and provide a direct comparison to emphasize ShapeMatch’s unique contributions?
>
> #### **AQ5:** Thank you for the question. While ShapeMatch follows the general consistency-regularization template of FixMatch (weak/strong augmentations with pseudo-labeling), it differs in two core aspects:
>
> #### - Instead of relying solely on early, often unreliable pseudo-labels from a deep model under label scarcity, ShapeMatch uses a shapelet model, robust in low-label regimes, to guide the Transformer/CNN during early training by matching its predictions.
> #### - We are also the first to use shapelet distances to refine and optimize pseudo-labels.
> #### - FixMatch uses generic data augmentations, whereas MTSC lacks domain-specific counterparts. ShapeAug leverages shapelets to preserve essential temporal patterns while perturbing less important regions, yielding diverse yet semantically faithful augmented series.
>
> #### These two components make ShapeMatch more robust and effective than a direct FixMatch adaptation for MTSC, especially in label-scarce settings.

---

### Official Review · Reviewer_jpnw · 2025-11-01

**Soundness:** 2
**Presentation:** 2
**Contribution:** 2
**Rating:** 2
**Confidence:** 4

**Summary:**

1.Proposes ShapeMatch: a novel semi-supervised framework that integrates classic Shapelet models with modern deep learning backbones.
2.Introduces Shapelet-guided training: leverages the robustness of Shapelet models in early training to guide deep models, providing strong inductive bias under low-label conditions.
3.Designs ShapeAug: a tailored augmentation strategy that identifies and preserves class-discriminative subsequences (shapelets) while applying controlled noise, masking, scaling, and shifting elsewhere.

**Strengths:**

1. The paper integrates two major MTSC techniques—shapelet and deep learning (DL) methods—by fusing them together, where the shapelet method essentially serves as a preprocessing or augmentation step for time series, followed by training with various DL backbones.
2. The experimental results appear strong, especially in demonstrating that ShapeMatch can be effectively applied to other DL methods as well.

**Weaknesses:**

1. Shapelet discovery is used as a preprocessing step because directly applying the shapelet method is computationally expensive despite its high accuracy, while deep learning methods offer a faster but less accurate alternative. However, this design does not truly combine the strengths of both approaches.
2. The approaches for shapelet discovery, shapelet distance feature definition, and training are not novel, resulting in a lack of innovation.
3. Weak and strong augmentation strategies have already been employed in contrastive learning for time series. The paper should cite relevant prior work, clearly highlight the differences, and include comparisons with similar methods in the experiments.
4. The decision not to augment the key subsequences is debatable; it may negatively impact the model’s generalization ability on most datasets.

**Questions:**

1. Using shapelet discovery as preprocessing—why not directly use the shapelet method? The main drawback of shapelet methods is their high computational cost despite high accuracy, whereas deep learning (DL) methods exhibit the opposite trade-off.
2. If it is useful, it should not be limited to semi-supervised scenarios—have you tried it in other settings?

---

> ### Author Response · Authors · 2025-11-27
>
> > **W1 & Q1:** Using shapelet discovery as preprocessing—why not directly use the shapelet method? The main drawback of shapelet methods is their high computational cost despite high accuracy, whereas deep learning (DL) methods exhibit the opposite trade-off.
>
> #### **AW1 & Q1:** Thank you for your question. In fact, our shapelets are also learnable. The key difference between our approach and the deep learning–based shapelet methods you mentioned is that we incorporate an offline phase to discover initial shapelets, which are then further optimized during training. This two-stage strategy has been shown to yield better performance in several recent works [1,2]. Moreover, if we remove the initial discovery step and instead randomly initialize the shapelets, we observe a performance drop of about 2–3%. Finally, our shapelet formulation eliminates the need to manually design the shapelet length, which is one of the most challenging hyperparameters to tune in [1,2].
>
> #### To quantify its effect, we compare ShapeMatch with and without the discovery step. Replacing discovery by random initialization consistently leads to 1.3–2.0% accuracy drops across both datasets and all label ratios (Table below), showing that the discovery phase is crucial rather than cosmetic. Further details are provided in Section F.4 of the revised appendix.
>
>
> | Dataset                               | APAVA (76.30%)  |            |            | FaceDetection (63.25%) |            |            |
> |---------------------------------------|------------------|------------|------------|-------------------------|------------|------------|
> | **Label Ratio**                       | **1%**           | **5%**     | **20%**    | **1%**                  | **5%**     | **20%**    |
> | **ShapeMatch + Shapelet Discovery**   | **60.26**        | **69.12**  | **74.11**  | **47.35**               | **55.13**  | **58.29**  |
> | ShapeMatch + Random Initialization    | 58.41            | 67.82      | 72.61      | 45.51                   | 53.15      | 56.62      |
>
> ---
>
> > **W2:** The approaches for shapelet discovery, shapelet distance feature definition, and training are not novel, resulting in a lack of innovation.
>
> #### **AW2:** We agree that the basic primitives, shapelet discovery, distance-based features, and supervised training, have been studied previously. However, our contribution does not lie in proposing a new shapelet algorithm, but in how these components are designed for semi-supervised, label-scarce time-series classification. Specifically, to the best of our knowledge, **ShapeMatch is the first framework** that (i) uses a shapelet model to guide a larger Transformer/CNN backbone in a semi-supervised setting, and (ii) introduces shapelet-based augmentation techniques tailored to this regime.
>
> ---
>
> > **W3:** Weak and strong augmentation strategies have already been employed in contrastive learning for time series. The paper should cite relevant prior work, clearly highlight the differences, and include comparisons with similar methods in the experiments.
>
> #### **AW3:** We appreciate this comment. Our weak/strong augmentation strategy is conceptually different from the techniques you mentioned, as we use shapelets to discover important local patterns and explicitly preserve them during the augmentation process. This ensures that the augmented time series remain diverse while still retaining the critical shape information.
>
> ---
>
> > **W4:** The decision not to augment the key subsequences is debatable; it may negatively impact the model’s generalization ability on most datasets.
>
> #### **AW4:** Our choice to avoid strong augmentation on the key subsequences is motivated by preserving the discriminative local shapes that the shapelets are designed to capture. Empirically, we observe that heavily distorting these key segments can degrade performance, as it blurs the very patterns the model should rely on. Instead, we apply stronger augmentations outside the key subsequences and use weaker transformations (e.g., slight jitter or scaling) where necessary to maintain label consistency. **An ablation in Table 4** demonstrates that aggressive augmentation on key segments (without ShapeAug) tends to hurt performance, whereas our current strategy (with ShapeAug) offers a better accuracy–robustness trade-off.
>
> ---
>
> > **Q2:** If it is useful, it should not be limited to semi-supervised scenarios—have you tried it in other settings?
>
> #### **AQ2:** In this paper, we primarily focus on the semi-supervised setting. However, we believe that the proposed techniques are useful in several other scenarios such as full supervised classification or anomaly detection as well, making them a promising direction for future work.

---

### Official Review · Reviewer_GzNv · 2025-11-01

**Soundness:** 2
**Presentation:** 2
**Contribution:** 2
**Rating:** 4
**Confidence:** 2

**Summary:**

ShapeMatch tackles semi-supervised multivariate time-series classification (MTSC) when labels are scarce. It first extracts class-specific shapelets from the labelled subset with a perceptually-aware algorithm (PPSD), trains a lightweight Shapelet Model and any deep backbone on labelled data, then continues semi-supervised training with “ShapeAug” augmentations and pseudo-labels that fuse frozen shapelet and deep-model predictions. Across 12 healthcare/UEA benchmarks and five backbones the framework improves 1 %–20 % label-regime accuracy over FixMatch, Semiformer, etc., while adding <1 h CPU shapelet discovery and 0.5 GB GPU memory.

**Strengths:**

1) Novel hybrid guidance mechanism: Combines shapelet distance features with deep logits for pseudo-labeling via epoch-scheduled and class-distance biases, yielding consistent +2–6 % accuracy over FixMatch on 12 datasets. Demonstrates complementarity: shapelets excel early when labels are extremely scarce, deep model dominates later, justifying the proposed curriculum.

2) ShapeAug augmentation strategy: Introduces shapelet-masked jitter/mask/scale/shift that preserves discriminative subsequences; ablation shows joint techniques outperform individual ones by up to 2 %.

3) Broad backbone compatibility: Evaluated with five architectures: TSLANet, iTransformer, ShapeFormer, MedFormer, PatchTST and a small CNN; ShapeMatch beats respective SSL baselines in every case, indicating framework generality.

**Weaknesses:**

1) Incomplete baseline coverage: Omits recent time-series contrastive or self-supervised methods that also work in label-scarce regimes; comparisons are limited to FixMatch, Pseudo-Label, Semiformer.

2) Scalability & computational bottleneck: Shapelet discovery remains CPU-bound (36 min on 8 cores for moderate data, Table 7); no GPU acceleration or complexity analysis with respect to V, T, or pool size g.

3) Limited analysis of failure modes: No dataset is identified where ShapeMatch underperforms supervised or SSL baselines; absence may indicate selective reporting.

4) Expand baseline coverage: Include at least two recent time-series SSL baselines under identical 1 %/5 %/20 % splits and report mean ± std to situate ShapeMatch novelty.

**Questions:**

See above weakness.

---

> ### Author Response · Authors · 2025-11-27
>
> > **Q1:** Incomplete baseline coverage: Omits recent time-series contrastive or self-supervised methods that also work in label-scarce regimes; comparisons are limited to FixMatch, Pseudo-Label, Semiformer.
> > **Q4:** Expand baseline coverage: Include at least two recent time-series SSL baselines under identical 1 %/5 %/20 % splits and report mean ± std to situate ShapeMatch novelty.
>
> #### **AQ1 & AQ4:** In the revised paper, we have additionally included two self-supervised methods (TS2VEC and CA-TCC) and one semi-supervised model (semiHGR) to further clarify the benefits of our approach. **As shown in Tables 1 and 2 in revised paper**, ShapeMatch consistently outperforms all these baselines across all comparison settings. We attribute this improvement to the shapelet guidance applied in the early training stage, which helps the backbone network learn more effective representations from the outset, thereby substantiating our contributions.
>
> ---
>
> > **Q2:** Scalability & computational bottleneck: Shapelet discovery remains CPU-bound (36 min on 8 cores for moderate data, Table 7); no GPU acceleration or complexity analysis with respect to \(V\), \(T\), or pool size \(g\).
>
> #### **AQ2:** We respectfully disagree that the shapelet discovery process presents a computational bottleneck. The 36 minutes reported correspond to using only 8 CPU cores; in practice, as shown in Table 7, our improved discovery procedure finishes in under one minute at a cost of only $0.368, without any performance drop.
>
> #### **Complexity Analysis:** The overall computational complexity of the shapelet discovery phase is based on the number of channels $V$, time-series length $T$, number of time series $N$, and number of selected shapelets $g$. This phase consists of two stages: *shapelet candidate discovery* and *shapelet selection*. In the candidate discovery stage, we search approximately $\mathcal{O}(0.2 N V T)$ salient points to obtain $g$ candidates. In the selection stage, we compute the information gain of each candidate over all $N$ time series, giving a complexity of $\mathcal{O}(g N^{2} D)$, where $D$ is the cost of computing the distance between one shapelet and one time series (with $D = T$ when using PISD as the distance). Summarising both stages, the overall complexity is $\mathcal{O}\bigl(0.2 N V T + g N T\bigr).$
>
> ---
>
> > **Q3:** Limited analysis of failure modes: No dataset is identified where ShapeMatch underperforms supervised or SSL baselines; absence may indicate selective reporting.
>
> #### **AQ3:** We did not selectively report datasets where ShapeMatch performs well. Our experiments include all five available medical time-series datasets and all UEA datasets with more than 500 training instances. The 500-instance threshold ensures that, even at the most challenging 1% label ratio, there are at least five labeled samples for training; below this, the semi-supervised setting becomes ill-posed and unstable.
>
> #### Within this pool of datasets, we did not observe any case where ShapeMatch consistently underperforms the supervised or SSL baselines; we therefore view this as a strength of our method rather than a limitation.

---

### Meta-Review · Area_Chair_dhuB · 2025-12-24

**Summary:**

This paper presents ShapeMatch, a semi-supervised framework for multivariate time-series classification that combines classical shapelet-based inductive bias with deep representation learning. The method uses a staged training strategy that initializes learning under limited labeled supervision with shapelet guidance, along with a multivariate-specific augmentation scheme. Experimental results on public benchmarks show that ShapeMatch consistently outperforms selected approaches.

**Reviewer Concerns:**

Most reviewers raised concerns about the paper’s overall contribution, the computational cost of time-series shapelet discovery during training, and the incompleteness of the experimental baselines for multivariate semi-supervised time-series classification. For instance, reviewer GzNv noted that the comparisons are limited to FixMatch, Pseudo-Label, and Semiformer, while reviewer a8G6 pointed out that only four semi-supervised baselines are included in the evaluation. Reviewer FMa2 further remarked that the model lacks innovation in capturing inter-variable relationships, and reviewer JtMc observed conceptual similarities between the proposed framework and FixMatch. Additionally, reviewer jpnw indicated that the shapelet discovery component offers limited methodological novelty.

**Reviewer Scores:**

The initial overall scores from reviewers GzNv, jpnw, JtMc, FMa2, and a8G6 were 4, 2, 6, 4, and 4, with corresponding confidence scores of 2, 4, 4, 5, and 5. After reviewing the full rebuttal and the authors’ consolidated responses, none of the reviewers participated in further discussion. I expect that reviewer jpnw may increase the overall score from 2 to 4, while the remaining reviewers are likely to maintain their original evaluations.

The primary remaining concerns relate to the limited novelty of the shapelet discovery process, the lack of an advantage in terms of training and inference efficiency, and the absence of an in-depth analysis of inter-variable relationship modeling.

---

### Decision · Program_Chairs · 2026-01-26

Reject